ecology/behaviour

nest morphology, gas exchange, fungus-growing ants, wind-driven flow, thermal convection, hypercapnia

**Author for correspondence:**
Martin Bollazzi
e-mail: bollazzi@fagro.edu.uy

# Carbon dioxide levels and ventilation in *Acromyrmex* nests: significance and evolution of architectural innovations in leaf-cutting ants

## Martin Bollazzi[1], Daniela Römer[1,2] and Flavio Roces[2]

[1]Entomología, Facultad de Agronomía, Universidad de la República, Av. Garzon 780, Montevideo 12900, Uruguay
[2]Department of Behavioral Physiology and Sociobiology, Biocenter, University of Würzburg, Am Hubland, Würzburg 97074, Germany

MB, 0000-0003-3361-5602; DR, 0000-0002-7437-2195; FR, 0000-0001-9258-3079

Leaf-cutting ant colonies largely differ in size, yet all consume $O_2$ and produce $CO_2$ in large amounts because of their underground fungus gardens. We have shown that in the *Acromyrmex* genus, three basic nest morphologies occur, and investigated the effects of architectural innovations on nest ventilation. We recognized (i) serial nests, similar to the ancestral type of the sister genus *Trachymyrmex*, with chambers excavated along a vertical tunnel connecting to the outside via a single opening, (ii) shallow nests, with one/few chambers extending shallowly with multiple connections to the outside, and (iii) thatched nests, with an above-ground fungus garden covered with plant material. Ventilation in shallow and thatched nests, but not in serial nests, occurred via wind-induced flows and thermal convection. $CO_2$ concentrations were below the values known to affect the respiration of the symbiotic fungus, indicating that shallow and thatched nests are not constrained by harmful $CO_2$ levels. Serial nests may be constrained depending on the soil $CO_2$ levels. We suggest that in *Acromyrmex*, selective pressures acting on temperature and humidity control led to nesting habits closer to or above the soil surface and to the evolution of architectural innovations that improved gas exchanges.

# 1. Introduction

Ant nests offer protection against predators and climatic extremes, yet they can compromise the air exchanges between the nest

inhabitants and the atmosphere. Especially in colonies of leaf-cutting ants, which cultivate a symbiotic fungus within subterranean chambers using plant fragments harvested by workers [1]. Due to the respiration of workers and fungus, considerable amounts of $O_2$ are consumed and $CO_2$ produced inside the nest chambers. All chambers of leaf-cutting ant nests exchange respiratory gases with the soil phase via diffusive flows, which strongly depend on the porosity of the soil. The direction of such an exchange would depend on the concentration of $CO_2$ in the soil relative to the concentration inside the chambers. When $CO_2$ levels in the soil are lower than inside chambers housing fungus gardens, $CO_2$ moves into the soil phase, yet diffusion would tend to be reversed at increasing nest depths, since levels of $CO_2$ in the soil phase increase with depth [2]. In the long term, it results in a dynamic equilibrium that equals the concentrations of $CO_2$ and $O_2$ inside a fungus chamber with those of the local surrounding soil, as far as no other ventilatory mechanisms driving the exchange of air between fungus chambers and the atmosphere are involved [3]. Thus, the concentration of $CO_2$ inside fungus chambers should be positively related to nest depth, as measured in field nests of *Atta* leaf-cutting ants [3]. Since high $CO_2$ concentrations are known to negatively influence the respiration rate of the *Atta* symbiotic fungus at values near 5% [4], both short-term and long-term responses to avoid increased $CO_2$ levels are expected to occur. As a short-term response, *Acromyrmex lundii* workers are known to relocate brood and fungus to nest chambers with $CO_2$ levels ranging from 1 to 3% [5,6]. Workers also use $CO_2$ as an orientation cue during nest excavation, avoiding levels of 4% and preferring places with 1% $CO_2$ for digging [7], which in the long term would result in the location of the colony at suitable $CO_2$ levels across the soil profile. Besides nest depth, it is known that external architectural features, such as the mound shape or the construction of nest turrets might promote gas exchanges through passive, wind-induced nest ventilation mechanisms [3,8–10].

Beyond the influence of the nest depth on the $CO_2$ levels inside the fungus chambers, it is an open question whether the underground nest architectures of the different leaf-cutting ant species, i.e. the arrangement of the nest chambers and their connections to the outside, evolved to facilitate the ventilatory air exchanges with the outside environment. Nests of leaf-cutting ants are initiated with the excavation of a single founding chamber by the mated queen [11], and the architecture of the adult nest is species specific and depends on colony size. The three known genera of leaf-cutting ants, *Atta*, *Acromyrmex* and *Amoimyrmex*, originated between 19 and 18 Ma ago, while the genus *Trachymyrmex*, the sister group of all of them that is a fungus grower yet not a leaf-cutter, shares a common ancestor with the leaf-cutting ants that evolved between 31 and 27 Ma ago [12–16]. To date, there is no conclusive evidence about the divergence time among the three leaf-cutting ant genera, nor an exhaustive phylogeny of *Acromyrmex* [17]. The transition from a non-leaf-cutting ant, *Trachymyrmex*, to *Atta*, *Acromyrmex* and *Amoimyrmex* leaf-cutting ants implied a huge increase in worker numbers and nest size to accommodate them, with a concomitant increase in the number of fungus chambers to be excavated, and with potential constraints for adequate air exchanges because of increasing colony size. *Trachymyrmex* colonies contain a few hundred workers and only some small-sized chambers excavated in a series along with a single vertical tunnel, which opens at the soil surface level [18]. *Acromyrmex* nests are generally many times larger than *Trachymyrmex* nests [1], yet considerably smaller than those of *Atta*. Adult *Atta* colonies are the largest among all ants, possessing several millions of individuals inhabiting giant subterranean nests with up to 7900 fungus chambers. Above ground, up to 100 nest openings are located in several large mounds or in one giant mound up to 10 m in diameter [19–21] (figure 1).

Regarding the underground architecture, nests of the basal *Trachymyrmex* grow from the founding chamber downwards by serially adding chambers along with a single, blind-ending vertical tunnel [22]. Chambers may be excavated by widening the vertical tunnel, thus giving the impression that the vertical tunnel runs through the chambers, or adjacent and connected to the tunnel via a single narrow tunnel called a peduncle, a feature observed in most higher Attina species that may have evolved to avoid delays in the provisioning of fungus gardens located downstream along with nest tunnels, and also to protect the fungus from direct exposure to circulating air (figure 1) [3,18,20,23–25]. It is important to indicate that in most basal taxa of fungus-growing ants, comprising species with comparatively small colony sizes, nest architectures are extremely diverse but appear to be much simpler, ranging from relatively exposed fungus gardens maintained under leaf litter, small nest chambers excavated or located in natural cavities, to arboreal nests with fungus gardens covered by accreted soil [26,27]. The serial pattern exhibited by *Trachymyrmex*, the sister genus of all leaf-cutting ants, can therefore be considered the basal architectural type from which the different nest architectures observed in all leaf-cutting ants evolved. For instance, such a serial chamber arrangement as observed in *Trachymyrmex*, single or multiplied in several branched tunnels, also

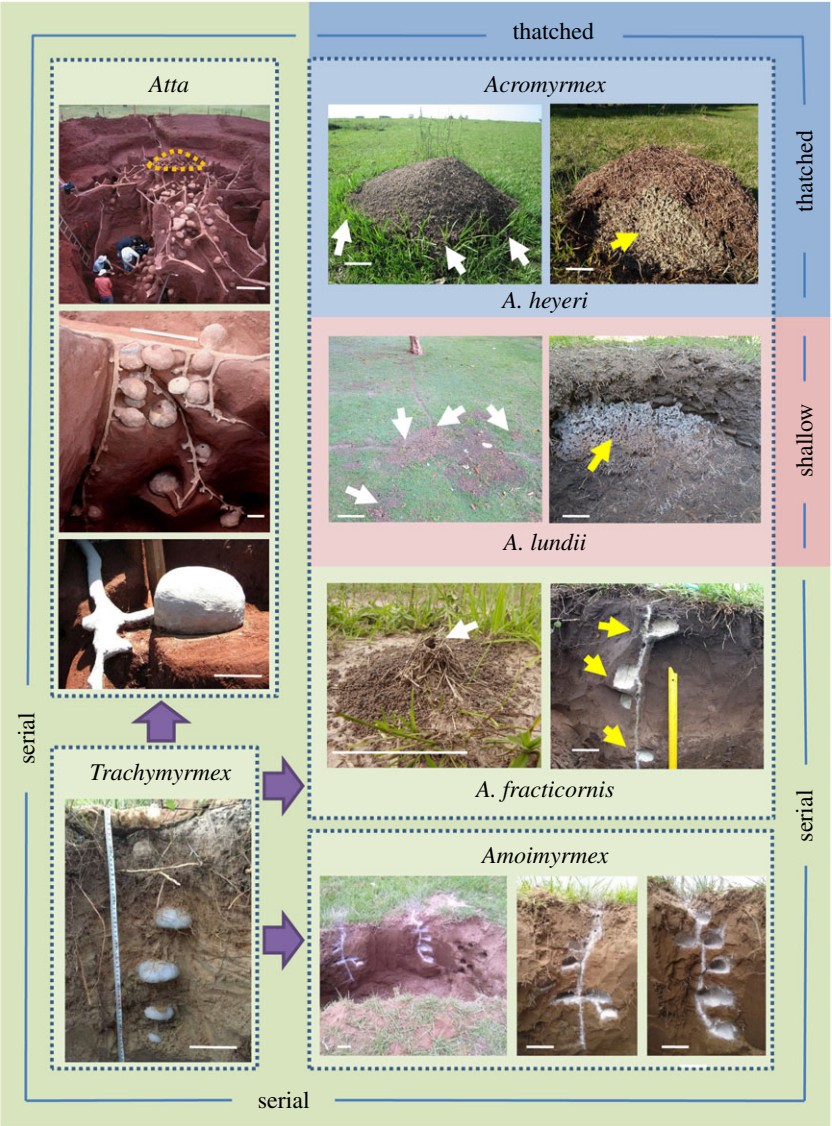

**Figure 1.** Representative nest types of the leaf-cutting ant genera *Amoimyrmex*, *Acromyrmex* and *Atta*, as well as their sister genus *Trachymyrmex*. **Trachymyrmex** nest: cast of a serial nest of *T. holmgreni* in southern Brazil, showing five chambers connected to the soil surface by a single blind-ending tunnel (nest cast and picture kindly provided by M.P. Cristiano). **Atta** nest: cast of a serial nest of *A. laevigata* (top) showing the serially connected chambers (middle) attached by a single short peduncle to a single tunnel (bottom). The area marked with the dashed line represents the location of the single external mound (pictures by L. Forti, reproduced from [3], with permission). **Amoimyrmex** nest: excavation of an *A. striatus* nest (left) showing two serial arrangements of fungus chambers marked with talc powder after fungus garden removal (middle and right). No underground connection between both arrangements was found (Rivera, Uruguay, pictures by Martin Bollazzi). **Acromyrmex** serial nests: *A. fracticornis* nest (partial view) showing its multiple chamber system (right) connected by one tunnel ending in a single entrance with a turret (left). Fungus gardens were removed and the chambers marked with talc powder (Formosa, Argentina, pictures by Flavio Roces). **Acromyrmex** shallow nests: *A. lundii* nest showing its single fungus garden (right) connected by many tunnels to the outside (left) (Montevideo, Uruguay, pictures by Martin Bollazzi). **Acromyrmex** thatched nests: *A. heyeri* nests showing the single fungus chamber (right) with its mound-shaped thatch made of grass fragments (left) (Tacuarembo, Uruguay, pictures by Martin Bollazzi). Yellow arrows indicate the fungus chambers, and white arrows the nest entrances. See table 1 for detailed information on the supporting bibliography. All scale bars represent 10 cm, unless for the *Atta* upper picture, which correspond to 1 m.

occurs in all species of the genera *Amoimyrmex* [15], *Atta* [3], and at least seven *Acromyrmex* species [25,28–30]. In these groups, colonies grow by extending both deeper and laterally from the location of the founding chamber, and the excavated chambers always branch off the connecting tunnel via a single peduncle (figure 1). The genus *Acromyrmex*, with at least 22 recognized

**Table 1.** Descriptions of different nest types as recovered from the bibliography for the 22 surveyed *Acromyrmex* species, the genera *Atta* and *Amoimyrmex*, as well as their sister genus *Trachymyrmex*. The *Acromyrmex* genus is represented as divided in the two still valid subgenera *Acromyrmex* and *Moellerius* [32]. However, evidence indicates that *A. heyeri* belongs to the *Acromyrmex* and not to the *Moellerius* subgenus [13,33] (see text and figure 1 for further explanations on nest types).

| taxa | | nest type | | |
| genus, species | | serial | shallow | thatched |
| --- | --- | --- | --- | --- |
| Acromyrmex (Acromyrmex) | A. ambiguus | | [29,30,34,35] | [36–39] |
| | A. aspersus | | [30,34] | [30,40] |
| | A. coronatus | | [30,38] | [38,41] |
| | A. crassispinus | | [30] | [30,34,38,42] |
| | A. diasi | | | [34,43] |
| | A. disciger | | [34] | [30,38,44] |
| | A. echinatior | | [45] | |
| | A. heyeri* | | [38,39,46,47] | [29,39,46–48] |
| | A. hispidus | [30] | | [29] |
| | A. histryx | | | [22,30,49,50] |
| | A. laticeps | | [30,34] | |
| | A. lundii | | [29,30,38,51] | [29,30,52] |
| | A. lobicornis | | [38,53–55] | [38,56,57] |
| | A. niger | [30,34] | | |
| | A. octospinosus | | [30,50,58] | [22,30,49,50] |
| | A. rugosus | [30,34,38,59,60] | | |
| | A. subterraneus | | [34,38,61,62] | [34] |
| | A. volcanus | | | [63] |
| Acromyrmex (Moellerius) | A. balzani | [28,30,34] | | |
| | A. fracticornis | [29] | | |
| | A. landolti | [25,50,64–66] | | |
| | A. versicolor | [67,68] | | |
| Atta spp. | | [19,21,69–73] | | |
| Amoimyrmex spp. | | [29,30,52,53,74] | | |
| Trachymyrmex | | [18,22,24,27,68,75] | | |

species [16], shows in addition to the serial nest type a variety of nest morphologies and architectural innovations not found in other leaf-cutting ants [31] (table 1). Architectural innovations in *Acromyrmex* as compared with other leaf-cutting ant genera refer to the way chambers are connected to the outside, their number and whether new chambers are excavated at deeper soil layers as the colony grows. We propose a distinction between three basic different nest types depending on both the depth at which fungus chambers are located and the degree of their connectivity to the outside (figure 1 and table 1).

First, the serial nests in which up to 30 chambers [59] containing the fungus gardens are connected via a single peduncle to a vertical tunnel that leads to the outside through a single opening. They can be uniserial nests as in the basal *Trachymyrmex* and several other fungus-growing ants genera [27,76], or multiserial, i.e. consisting of a group of connected uniserial arrangements that lead to the outside via several openings, as in *Amoimyrmex* [74] and *Atta* [3,19,21,69]. Besides, nest entrances may have a mound of loose soil around them, or a turret resulting from the import and arrangement of materials [77,78]. This basal architectural type of leaf-cutting ants is exemplified by the nests of *Acromyrmex fracticornis* (figure 1). Because of diffusive gas exchanges with the soil,

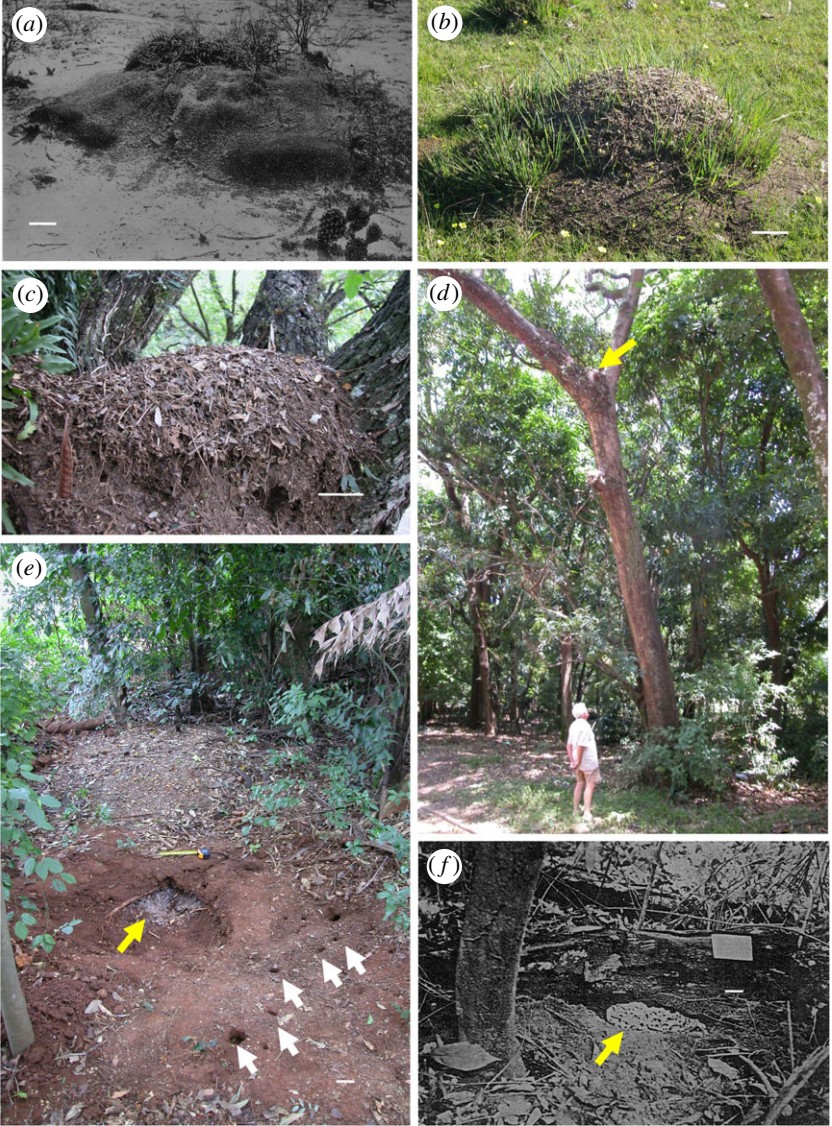

**Figure 2.** Examples of *Acromyrmex* species inhabiting shallow and thatched nests. (*a*) First published picture of a South American *Acromyrmex* nest showing the external architecture of a shallow *Acromyrmex lobicornis* nest in northwest Argentina (picture by Carlos Bruch [53]). (*b*) A nest of the same species, *Acromyrmex lobicornis*, from south Uruguay, clearly showing a thatch made of leaf fragments, as opposed to the sallow nesting habit shown in (*a*) (picture by Martin Bollazzi). (*c*) An *Acromyrmex coronatus* thatched nest built on the angle formed by two branches of a tree in Piracicaba, Brazil (picture by Martin Bollazzi). (*d*) Another *A. coronatus* thatched nest photographed in the same location as in (*c*), showing the height at which nests could be built (yellow arrow; picture by Flavio Roces). (*e*) A partially excavated shallow nest of *Acromyrmex subterraneus* exposing one fungus garden (yellow arrow). The several nest entrances (white arrows) were exposed after cleaning the forest floor from leaf litter (Piracicaba, Brazil, picture by Martin Bollazzi). (*f*) The single fungus garden of a shallow nest of *A. octospinosus* in Galeota, Trinidad and Tobago (picture taken by Neil Weber, from [79]). All scale bars represent 10 cm.

$CO_2$ levels in the fungus gardens are expected to be in a dynamic equilibrium with those of the adjacent soil, unless other ventilatory mechanisms enhance exchanges of air between the nest and the atmosphere.

The second nest type, the shallow nest, is composed of either one, much larger or several shallow chambers located at approximately the same depth, being connected to the outside by more than one tunnel, as in *Acromyrmex lundii* (figure 1). At least 12 *Acromyrmex* species have been reported having shallow nests (table 1 and figure 2*e,f*). This architectural innovation of increasing nest size while keeping chambers at a shallow depth seems to result from the expansion of the nest at the same depth at which the queen dug the founding chamber. *Acromyrmex* queens have been reported to dig founding chambers at 5–10 cm depth [48,61,80,81], shallower than the *ca* 20 cm excavated by queens

of a number of *Atta* species [11]. Externally, the several entrances may have mounds of loose soil from the nest excavation surrounding the openings [29,30].

Colonies belonging to a third group of *Acromyrmex* species build thatched nests (table 1), as in *Acromyrmex heyeri* (figure 1). Usually, a single fungus garden is located above ground at the soil surface level, and it is covered by a thatch made of leaf fragments foraged for building purposes, along with collected dry plant fragments (electronic supplementary material, S1) and excavated soil to a lesser extent. *Sensu lato*, thatched nests comprise all cases in which a single or several fungus gardens are placed at the soil level, or completely above the soil, and covered by a thatch, being built either at the base of shrubs and trees, or on the bare ground [49,56]. Numerous *Acromyrmex* species have been reported as inhabiting both nest types, shallow and thatched [31] (figure 2a,b). Such variability seems to be linked to different habitats and in only the case of *A. heyeri*, sympatric occurrence of both nest types has been extensively studied [46,47]. Compared with the serial and shallow, the thatched nest offers the highest degree of connectivity to the outside because of its porous nature, which resembles that of the organic mulch used in agriculture. Mulches are well known for allowing air to move freely between the soil and the atmosphere [2]. It is quite remarkable that the fungus garden is almost completely placed above the soil surface level, and that workers import building materials to construct an external self-supporting structure that protects the entire fungus and functions as the nest (electronic supplementary material, S2, video). Hence, the thatch represents an innovation in building behaviour as compared with the deposition of excavated soil, and sometimes of other imported materials, in the form of mounds and turrets around nest entrances, which do not contain fungus gardens. Remarkably, colonies of some thatching *Acromyrmex* species even left completely behind the habit of nesting in the soil and build thatched nests at the base, or even the crowns of trees [1,30,82] (figure 2c,d). To our knowledge, *Acromyrmex* is the only South American ant genus in which thatched nests occur in several species throughout the continent. Indeed, it parallels the thatching nesting habits of the *Formica* genus in the Northern Hemisphere, both in species number exhibiting the nesting habit and in its continental distribution [83].

What were the driving forces behind the evolution of architectural innovations in the genus *Acromyrmex*? While the occurrence of subterranean and superficial nesting habits may have been favoured during evolution because of thermoregulatory needs [31], it is unclear whether the control of other environmental variables was also under selection. In general terms, it is reasonable to hypothesize that a set of different interacting selective forces, instead of a single one, led to the evolution of species-specific nest architectural traits [84], for instance those favouring the control of climate variables inside the nest, i.e. temperature, humidity and air exchanges, or facilitating the protection against pathogens and predators, or the spatial organization of nest tasks [85–88]. Nonetheless, the control of the nest environment appears to be one of the main selective forces responsible for the evolution and plasticity in nest design. In nasute termites, for example, the evolutionary transition from arboreal nesting habits to mound-building appears to have been driven by drier environmental conditions, under which mounds could better retain moisture [89]. In a broad sense, it has been suggested that the architecture of termite mounds represents an adaptation to local environmental conditions [90]. In addition to species-specific differences in nesting habits, phenotypic plasticity in mound architecture has also been discussed in *Macrotermes bellicosus* as resulting from a trade-off between thermoregulatory needs and control of gas exchanges [91]. While a few cases of phenotypic plasticity in nest design are known in leaf-cutting ants (summarized in [31]), to our knowledge, there are no comparative studies on evolutionary trade-offs between traits related to the control of nest climate in ant nests. For leaf-cutting ant nests, we hypothesize that different trade-offs may occur, as follows: a serial arrangement of nest chambers as observed in *Amoimyrmex*, *Atta* and several *Acromyrmex* would probably provide benefits in terms of moisture and temperature control, because nests would be less exposed to environmental fluctuations, yet at the expense of reduced gas exchanges. Shallow and thatched nests, as observed in *Acromyrmex*, may on the contrary be more susceptible to desiccation [92,93], yet they may benefit in terms of both thermoregulation [46] and probably improved gas exchanges.

We therefore asked whether the architectural innovations in *Acromyrmex* nesting habits were selected to facilitate ventilatory gas exchanges as compared with the serial, basal nest arrangement. In the present study, we experimentally investigated the effect of the different nest architectures of *Acromyrmex* nests on their ventilation and $CO_2$ levels inside chambers. In the laboratory, we compared the $CO_2$ concentration in fungus gardens placed inside laboratory nests resembling the three different nest types found in *Acromyrmex*, i.e. serial, shallow and thatched, while controlling for variables that are known to affect

nest ventilation and $CO_2$ levels, such as varying soil porosity, wind and temperature. In the field, we comparatively assessed $CO_2$ concentrations inside nests of two *Acromyrmex* species inhabiting shallow and thatched nests, i.e. nests with architectural innovations, and explored different mechanisms potentially responsible for nest ventilation. Finally, the effects of architectural innovations on nest ventilation in *Acromyrmex* are discussed in an evolutionary framework.

# 2. Material and methods

## 2.1. Ventilation in *Acromyrmex* laboratory nests with different architectures

Laboratory experiments were designed to comparatively investigate the effects of wind on passive ventilation and $CO_2$ levels as depending on the different nest architectures. For that, we have compared the concentration of $CO_2$ inside fungus chambers arranged in the three nest types described above, i.e. nests with serial, shallow and thatched architectures (figure 1), with that measured inside tightly closed fungus chambers of the same size. All fungus chambers housed a subcolony of *Acromyrmex lundii*, i.e. chambers completely filled with fungus and colony members were detached from the main colony and organized to recreate the three nest architectures described below. Thus, the goal of the laboratory experiments was to assess the importance of wind for the ventilation of each fungus chamber as compared with a closed chamber, for each of the three nest architectures.

Nests with the three nest architectures were established in a rack consisting of a top-placed polycarbonate shelf (50 × 50 cm), acting as outside surface level, and underneath shelves that allowed the location of fungus chambers at four different depths. At the centre of each shelf, a 25 mm hole enabled the interconnection of the fungus chambers with the outside by a vertical transparent PVC tube.

The three nest architectures were arranged as follows. First, the serial nest was composed of four fungus chambers (transparent plastic boxes: 10 × 10 × 6 cm), each placed at a shelf located at 0.25, 0.5, 1 and 1.5 m depth (figure 3*a*, left). They were connected by a 5 cm peduncle of PVC tubing (10 mm Ø) to a vertical tube running from the deepest chamber upwards, which opened to the outside via a single nest entrance. Second, the shallow nest consisted of a single fungus chamber, of the same size used for the serial nests, located at 0.25 m depth and connected to the outside level by two tunnels starting at its sides (figure 3*b*, left). Third, the thatched nest was simulated by attaching a single fungus chamber equal in size to those used in the previous nests, without a lid directly beneath the shelf acting as outside surface level, via an aperture similar in size and shape to the top side of the chamber. The thatch was built by covering the upper side of the exposed fungus with leaf fragments previously cut by workers and stored in the foraging arena as a leaf cache (figure 3*c*, left). The thatch had a basal diameter of 15 cm and a maximal height of 10 cm, being roughly one-fourth the size of natural thatched nests of *A. heyeri* and other thatch-building *Acromyrmex* [46,56,94].

During measurements, $CO_2$ concentration inside fungus chambers was assessed as a function of wind speed by blowing air over the outside level at values of either 0, 1 and 2.5 m s$^{-1}$. Wind-induced ventilation was evinced by comparing the $CO_2$ levels in the same fungus chamber when the connections to the outside were closed, or the thatch tightly covered, so that no ventilation at all could occur. The contribution of diffusive gas exchanges through the nest openings was assessed by comparing levels in the open nest chambers under no-wind conditions (0 m s$^{-1}$) with those in the tightly closed chambers. It is important to emphasize that impermeable plastic boxes were used as nest chambers, to preclude diffusive flows through the chamber walls. Under natural conditions, diffusion of respiratory gases may occur through the soil structure, depending on several variables such as soil porosity, soil moisture, etc. Precluding such diffusive flows in our experiments allowed to address the ventilatory effects of wind-induced flows through the nest tunnels alone.

All assays were initiated by first pulling $CO_2$-rich air from inside the chambers with a peristaltic pump, allowing for inflow of room air, so as to reach near-atmospheric $CO_2$ levels of *ca* 0.10% inside the nests. After reaching that level, the pump was turned off and the assay started. Levels of $CO_2$ inside the fungus chambers were measured after 60 min. We decided to use this time interval based on preliminary experiments showing that without ventilation, $CO_2$ levels inside the fungus chamber reached values between 2 and 3% after 1 h. Values around 4% are known to be avoided by workers of the present species because of potential harmful effects on their fungus [5]. As a consequence, our measurements encompassed $CO_2$ concentrations below levels that are detrimental for the fungus garden. Additionally, a previous study in our group [4] showed that the production of $CO_2$ by an

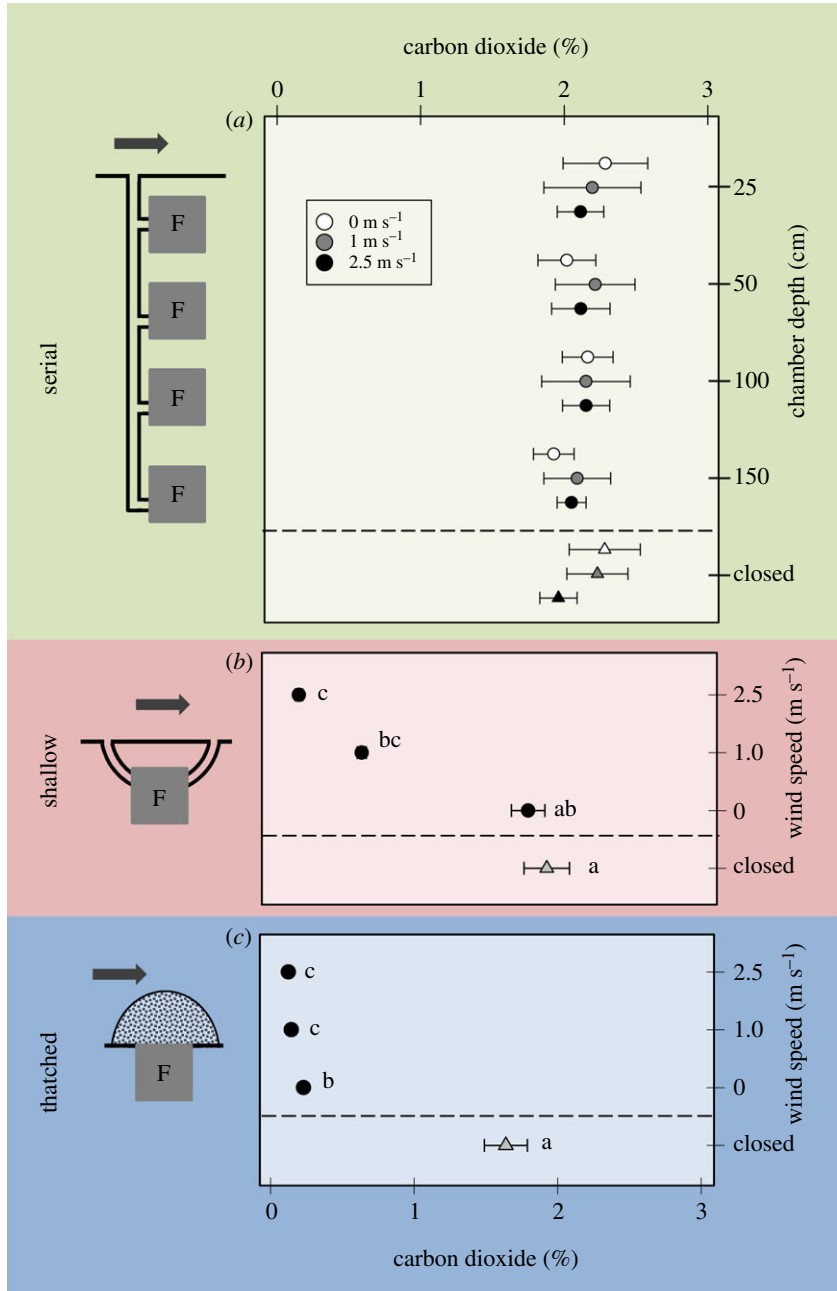

**Figure 3.** Carbon dioxide concentration (%, mean ± s.d., $n = 10$) as a function of depth and wind speed (black, grey and white circles) inside fungus chambers in laboratory nests representing the three different nest architectures found in the genus *Acromyrmex*. (*a*) Serial nest with only one tunnel connecting to the outside. (*b*) Shallow nest with a single chamber and two tunnels connecting to the outside. (*c*) Thatched nest. Left: schematic drawings of the different nest arrangements. Values are the $CO_2$ concentrations inside the fungus chambers measured after 1 h, starting from atmospheric values (see Material and methods for further details). The condition 'closed' (triangles) corresponded to tightly sealed fungus chambers without ventilation. F = chamber filled with fungus garden (see text for details). Values labelled with the same letter do not differ statistically (see text for statistics).

enclosed fungus garden of the leaf-cutting ant *Atta sexdens*, which cultivates the same fungus species of *A. lundii* [95], follows a linear increase over 60 min, reaching values up to 3%.

For measurements, each fungus chamber was connected to a $CO_2$ sensor with an attached voltage data logger (GS 20 ED/$CO_2$, Sensor Devices, Germany; Tinytag® data loggers) by a length of PVC tubing, acting as outflow, equipped with a peristaltic pump that returned the air into the chamber, creating hence a closed circuit with no pressure modification. For all three nest arrangements, wind was simulated by a fan placed at the surface level so as to direct the airflow 15 cm above nest

entrances or the thatch. In the series under wind conditions, the fan was turned on at the beginning of each assay and continued to generate wind over 60 min. Wind speed was adjusted at three values: 0, 1 and 2.5 m s$^{-1}$ by a voltage controller and controlled with a hot wire omnidirectional anemometer (TSI Instruments, Model 8475-300-1). Ten replicates for each wind speed were performed for both the shallow and thatched nests, and eight for the serial nests.

For the laboratory experiments, three *Acromyrmex lundii* colonies were collected in Tabaré, Florida, Uruguay (33°21′33.74″ S, 55°35′33.38″ W) and transported to the Department of Behavioral Physiology and Sociobiology at the University of Würzburg, Germany. Colonies were allowed to grow in 10 × 10 × 6 cm transparent plastic boxes connected by tubes of 1 cm internal diameter and maintained in a climatic room under 25°C and 12 L : 12 D cycle. By the time of the experiments, each colony was composed of *ca* 12 boxes filled with fungus gardens.

## 2.2. Effect of nest architecture on ventilation and CO$_2$ concentration in *Acromyrmex* field nests

To obtain a more comprehensive view of the effects of the nest architecture on nest ventilation under natural conditions, the laboratory measurements, as described above, were complemented with measurements on field colonies. In the field, we comparatively investigated ventilation mechanisms and CO$_2$ levels inside nests of the two sympatrically ocurring *Acromyrmex* species having nests with architectural innovations, i.e. *A. lundii* building shallow nests and *A. heyeri* building thatched nests. Regrettably, no measurements on serial nests were possible, since no sympatric *Acromyrmex* species with that nest architecture occurred at the study site. However, serial nests have been extensively investigated regarding ventilation mechanisms in the genus *Atta* [3,9], which may allow, despite their large size differences, general inferences about the effect of a serial arrangement of chambers on nest ventilation.

Field studies were conducted in Tabaré, Florida, Uruguay (33°21′33.74″ S, 55°35′33.38″ W). Nests were located in an intensive cattle-harvested prairie with no shrubs, dominated by prostrate grasses less than 5 cm in height. To comparatively assess CO$_2$ concentrations inside shallow *A. lundii* and thatched *A. heyeri* nests, a prospective probe attached to a CO$_2$ sensor was used. The probe consisted of a metal pipe of 8 mm internal diameter, which was gently introduced inside the fungus gardens, similarly to the procedures used for *Atta* nests in previous studies [3,4,96]. After the fungus chamber was reached, evidenced by the large number of alarmed ants climbing up the pipe, two parallel PVC tubes, 3 mm internal diameter, were gently introduced through the pipe until its end. These PVC tubes acted as inflow and outflow for a CO$_2$ sensor via a peristaltic pump, as described for the laboratory experiments, thus generating no pressure differences nor changes of the actual CO$_2$ levels inside the fungus chamber. A computer attached to the CO$_2$ sensor recorded the CO$_2$ concentrations. A total of 10 *A. heyeri* and eight *A. lundii* nests were measured. For statistical analysis, additional values obtained in simultaneous measurements of interspecific nest pairs (see below) were also included, making up a total of 14 and 12 nests, respectively. Measurements were carried out during daytime for at least 30 min in each nest.

In order to comparatively address the effect of wind blowing at the soil surface on CO$_2$ levels and nest ventilation, simultaneous measurements were performed in four interspecific nest pairs, i.e. one *A. heyeri* nest and one neighbouring *A. lundii* nest. For each nest of the pair, a prospective probe was gently introduced into the fungus garden as described above and attached to a single CO$_2$ sensor connected to a gas multiplexer that allowed to switch between the fungus chambers of the two colonies. A third gas switcher input was connected to the atmospheric air so as to clean the CO$_2$ sensor, the gas multiplexer and the tubing in between measurements. An omnidirectional anemometer (Thies Clima Model 4.3515.50.00, ±0.01 m s$^{-1}$) attached to a count data logger (Tinytag®), placed equidistant from both nests and at the same height of the nest mound of *A. heyeri*, was used to record wind speed. For each of the four nest pairs, simultaneous measurements were carried out exclusively on days with no rain and clear sky. Only four pairs could be used due to the requirements that nests had to occur in soils with no slope, identical vegetation cover and close enough to each other (less than 25 m) to make the measurements technically possible. Interval time between measurements of paired nests depended on the distance between them, being longer, the greater the distance, given that the sampled air has to travel through longer tubing. For those paired nests located far away, interval time between recorded CO$_2$ values reached several minutes, and given the high variability of wind speeds, it was difficult to assign a single wind value to both measurements. In order to obtain comparable values, CO$_2$ samples of both nests and wind measurements were averaged over 30 min for statistical analysis. Total

measuring time was limited by the onset of rains or technical issues. Measuring times achieved for the four nest pairs were 21, 8, 10 and 28 h.

Besides the relevance of wind, nest ventilation may also be passively driven by thermal convection. Because of their porous structure and being located above ground, i.e. more exposed to environmental fluctuations, thatched nests may rely more strongly on thermal convection than underground nests. To evaluate the contribution of thermal convection for the ventilation of thatched nests, field nests of A. heyeri were airtight sealed to control for the effects of wind, and the $CO_2$ concentrations in their fungus gardens were measured over 24 h before and after the experimental manipulation. Four A. heyeri nests were selected for the experiments, measurements being carried out all day long every 10 min. Throughout experiments, temperature inside the fungus and of the outside air were also recorded (Tinytag® data loggers): the temperature difference between the fungus garden and the external air was considered a measure of the ventilation potential driving mass movements of air from the nest interior to the environment. Carbon dioxide was measured inside the fungus garden as indicated above using a drilling probe attached to a peristaltic pump and a $CO_2$ sensor. Nests were airtight sealed by using a greenhouse plastic sheet and by placing sand at its edges around the mound base. Measurements of the four different nests were pooled, and carbon dioxide concentrations as a function of the temperature difference between fungus garden and outside were compared in both the airtight sealed and open nests.

# 3. Results

## 3.1. Ventilation in Acromyrmex laboratory nests with different architectures

The $CO_2$ levels measured inside the chambers of the laboratory nests strongly depended on their architecture. Inside a serial nest, $CO_2$ levels reached ca 2% after an hour, at all chamber depths (figure 3a). For the three wind speeds assayed, the $CO_2$ concentrations inside the fungus chambers did not differ from that of the closed, control chamber, indicating the lack of both diffusive and convective flows through the single nest entrance, and no wind-induced ventilation. In addition, there was no effect of chamber depth (two-way ANOVA-repeated measures following a reciprocal transformation, wind speed $F_{2,119} = 0.51$, $p = 0.61$, depth $F_{4,119} = 1.39$, $p = 0.26$, wind speed × depth $F_{8,119} = 2.1$, $p = 0.06$).

On the contrary, $CO_2$ concentrations inside a shallow nest, i.e. a single fungus chamber simultaneously connected to the outside through two tunnels (figure 3b), decreased with increasing wind speed over the nest surface (after unsuccessful variable transformation, Dunn's post hoc test following a Friedman repeated-measures ANOVA $X^2_3 = 28.08$, $p < 0.001$). While $CO_2$ concentration inside the closed chamber did not significantly differ from that of the no-wind situation (0 m s$^{-1}$), indicating the lack of both diffusive and convective flows through the nest entrances, a significant decrease in $CO_2$ levels with increasing wind speeds of 1 and 2.5 m s$^{-1}$ was recorded.

Thatched nests showed very low $CO_2$ levels (figure 3c), even under no-wind conditions, suggesting a strong involvement of diffusive and convective flows that promoted ventilation. When the wind was blowing over the thatch, $CO_2$ levels significantly decreased as compared with the no-wind conditions. Closing the nest to completely preclude ventilation led to a significant increase in the $CO_2$ levels, as also observed in the two nest arrangements described above. (Tukey's post hoc test after a one-way ANOVA repeated measures following a reciprocal transformation, $F_{3,39} = 398.14$, $p < 0.001$).

## 3.2. Effect of nest architecture on ventilation and $CO_2$ concentration in Acromyrmex field nests

In the field, the mean $CO_2$ levels inside shallow A. lundii nests, averaging 2.08%, were approximately twice as high as in thatched A. heyeri nests, with values of 1.16% (figure 4a) ($t_{24} = -4.65$, $p < 0.001$).

The simultaneous measurements on four paired A. lundii–A. heyeri nests allowed to evaluate the effect of natural winds blowing over the nest on the $CO_2$ concentrations inside the fungus chambers. Figure 4b shows an example of the simultaneous measurements of a nest pair, to illustrate the daily pattern of $CO_2$ levels as a function of wind speed. Inside the A. lundii shallow nest, $CO_2$ levels slightly varied over the measuring time of around 21 h, with values between 1.8% and 2.3%. Inside the A. heyeri thatched nest, on the contrary, $CO_2$ levels were much more variable and lower than those measured in the shallow nest (figure 4b).

To comparatively investigate the effect of wind speed on the $CO_2$ levels inside fungus gardens, it was necessary to standardize the data obtained from the four paired nests, because absolute $CO_2$ levels may depend for instance on unknown variables such as nest size and depth. For each separate nest, we calculated the per cent change in $CO_2$ levels as a function of wind speed, i.e. the difference between

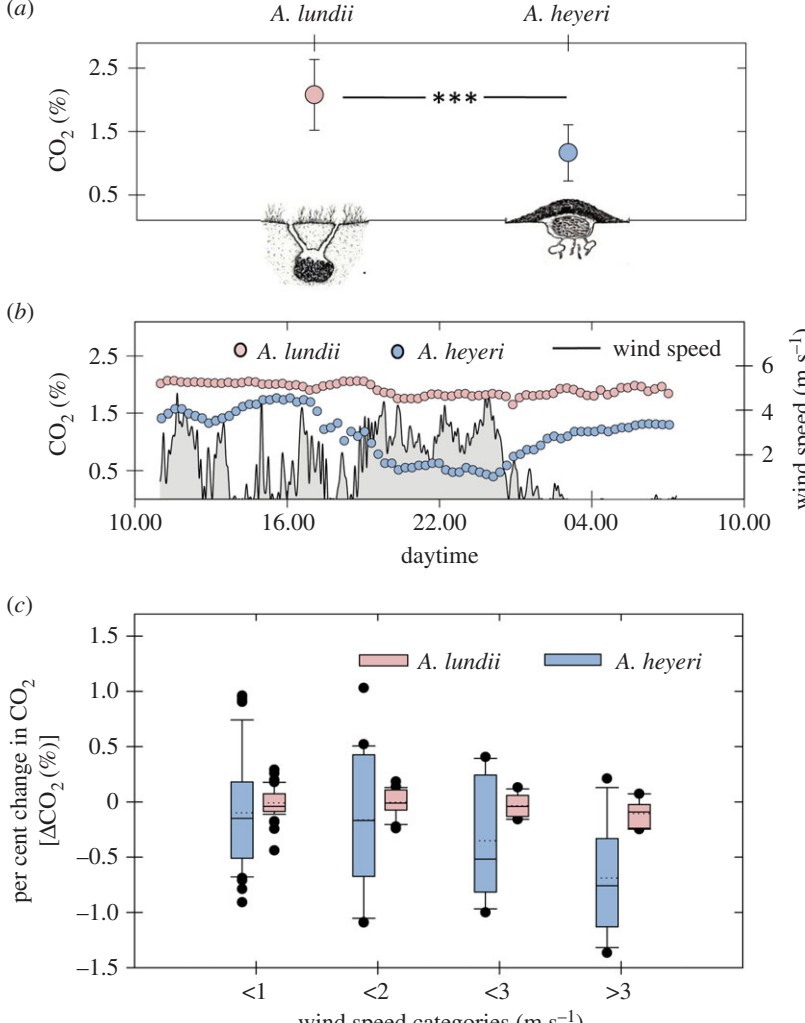

**Figure 4.** Comparative $CO_2$ concentrations in field nests of two *Acromyrmex* species inhabiting shallow and thatched nests. (*a*) Average $CO_2$ concentration (%, mean ± s.d.) inside the fungus gardens of *Acromyrmex lundii* shallow nests ($n = 12$) and *Acromyrmex heyeri* thatched nests ($n = 14$). *A. lundii* nest drawn by Zollessi [51]; *A. heyeri* nest drawn by Bonetto [29]. (*b*) Example of a simultaneous measurement of $CO_2$ inside fungus gardens of an *A. lundii–A. heyeri* nest pair, as well as wind speed, as a function of measuring time. (*c*) Per cent change in $CO_2$ levels as a function of wind speed categories for the pooled data of the four experimental *A. heyeri–A. lundii* nest pairs (see text for further details).

the $CO_2$ concentration for each measurement at a given wind speed minus the average concentration measured in total absence of wind for the nest under consideration. Values were expressed as a per cent decrease or increase in $CO_2$ levels, and correlated to wind speed by computing the repeated measures correlation coefficients ($r_{rm}$) [97,98] (package 'rmcorr' for R) with nest as subjects, yielding for *A. lundii* a coefficient $r_{rm} = -0.26$ ($p < 0.05$), and for *A. heyeri* $r_{rm} = -0.21$ ($p < 0.05$). It indicates that wind promoted a slight but significant reduction of $CO_2$ inside the fungus gardens of both nest types, shallow and thatched, and that comparatively, the thatched structure of the *A. heyeri* nests allowed wind-induced ventilation to occur much more strongly than in the shallow *A. lundii* nests. The low correlation coefficient can partially be explained by the fact that wind gusts did not induce a rapid decrease of $CO_2$ levels, since high and stable $CO_2$ concentrations were recorded despite high wind speeds over short periods of time. An example of high $CO_2$ concentrations in the *A. heyeri* nest, despite wind gusts, is presented in figure 4*b* around 16.00, which can be compared with the slight yet stable decrease observed at 22.00. It seems that a stable reduction in $CO_2$ levels is achieved only when high wind speeds occur over long periods of time, suggesting that the nest structure acts as a kind of low-pass filter for high-frequency changes in wind speed. Figure 4*c* presents the per cent change in $CO_2$ levels as a function of wind speed categories, for the pooled data of the four experimental *A. heyeri–A. lundii* nest pairs (figure 4*b*; electronic supplementary material, S3). It clearly shows that

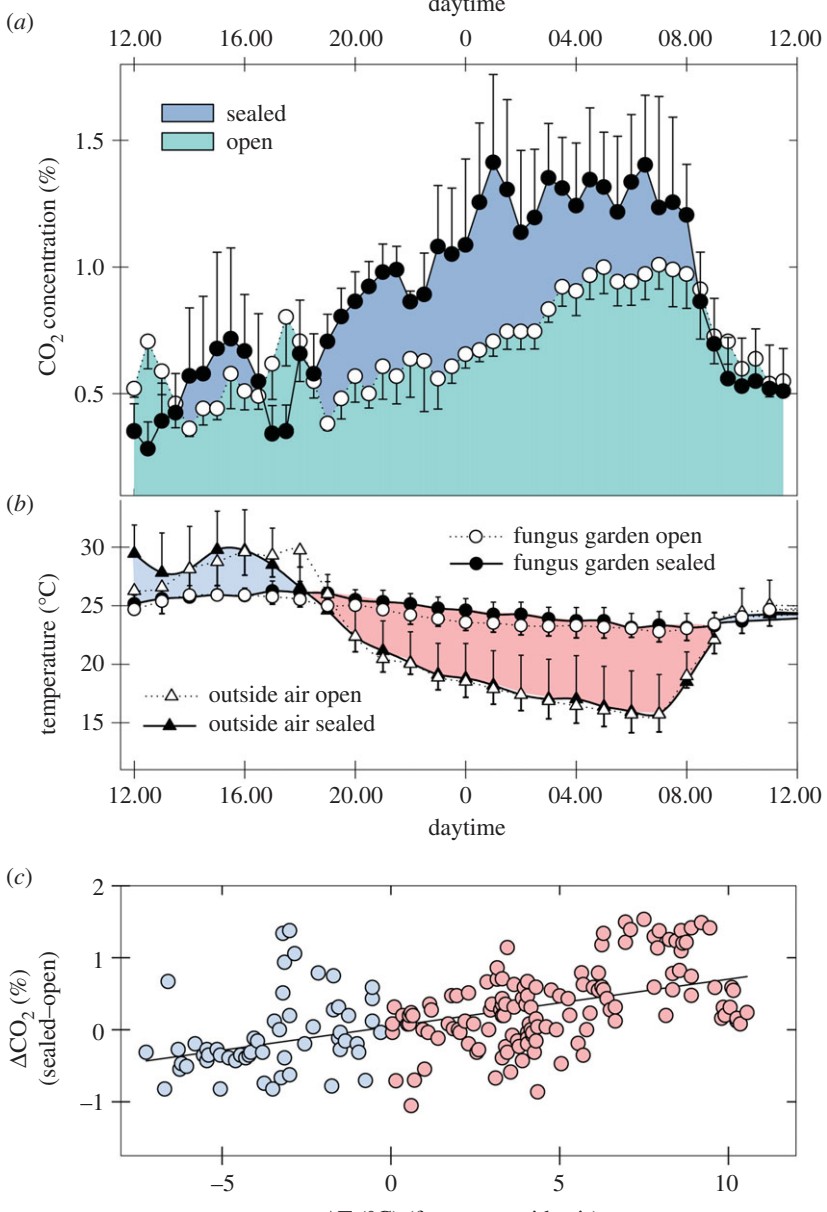

**Figure 5.** Daily pattern of $CO_2$ levels inside four field nests of *A. heyeri* before and after the thatch was sealed to preclude wind-induced ventilation. (*a*) $CO_2$ concentration inside fungus gardens as a function of daytime (mean ± s.d.), while nests remained open and after the thatch was sealed. (*b*) Simultaneous measurements of temperatures (mean ± s.d.) inside and outside the fungus garden, for both sealed and open nests. The blue marked area indicates a negative temperature difference between fungus and outside air, while the red area indicates a positive difference. (*c*) Difference in $CO_2$ concentrations between the airtight sealed and the open condition (from (*a*)) as a function of the thermal difference between fungus and outside air (average of black and white circles minus average of white and black triangles in (*b*)). Blue dots for negative differences and red dots for positive differences.

the per cent change of $CO_2$ is extremely variable for *A. heyeri* nests as compared with *A. lundi* nests at all wind speeds, and furthermore, that it could even reach positive values, i.e. an increase in wind speed led to an unexpected increase in $CO_2$ levels inside the nest, mostly at low and mid wind speeds. The reason for this phenomenon could be that wind draws $CO_2$-rich air from lower areas of the nest towards the fungus garden area where the sensor was located, and that $CO_2$ clearance was delayed or only occurred at high and sustained wind speeds.

Data from the field experiments aimed at evaluating the effect of thermal convection on $CO_2$ levels inside thatched nests are presented in figure 5, for nests before and after the airtight sealing of the

thatch. Over the 24 h of recordings, $CO_2$ levels increased at night in the open nests and much more strongly in the sealed nests (figure 5a). Simultaneous measurements of temperatures inside and outside the fungus garden, which may account for the observed daily pattern in $CO_2$ levels, are presented in figure 5b. While fungus temperature remained almost stable around 24°C, for both open and sealed nests, outside temperature followed the natural daily pattern and varied between a minimum of approximately 15°C close before sunrise, and a maximum of 30°C in the afternoon. To emphasize the importance of thermal convection as a driver of mass movements of air from the nest interior to the environment, we calculated the temperature differences between the open and sealed nests over the 24 h period and their correlation with the corresponding $CO_2$ levels.

The temperature difference between fungus and outside air showed a marked daily dynamic (figure 5b), which positively correlated with the concentration of $CO_2$ in both the sealed and open nests: the larger the thermal difference, the higher the $CO_2$ levels measured (repeated measures correlation coefficients with nests as subject; sealed nest: $r_{rm} = 0.71$, $p < 0.001$; open nest: $r_{rm} = 0.52$, $p < 0.001$). The effect of such thermal difference on $CO_2$ levels was much stronger for the sealed than for the open nests, i.e. sealed nests have much higher $CO_2$ levels than open nests, the higher the thermal differences between nest interior and outside air (repeated measures correlation coefficient with nests as subject: $r_{rm} = 0.50$, $p < 0.001$ (figure 5c).

# 4. Discussion

Taken together, our experimental results showed that levels of $CO_2$ inside fungus gardens of *Acromyrmex* nests are strongly dependent on their architecture, the nest connectivity to the outside, and particularly the presence of a thatched mound, which can be regarded as the key architectural innovation that enabled improved gas exchanges by taking advantage of different ventilation mechanisms.

## 4.1. Ventilation mechanisms and $CO_2$ levels inside *Acromyrmex* nests

When comparing the three nest types in the laboratory, results showed that in the ancestral, serial nest type with a single nest opening, wind-induced flows was not involved. Wind-induced flows require that air circulates through a tunnel with two openings connecting to the outside, one acting as inflow and the second as outflow opening. In serial nests, both uni- or multiserial tunnels running downwards, to which chambers are attached, end blind, thus precluding air circulation. As a consequence, air exchanges in a nest with a serial architecture are expected to occur via diffusion in the surrounding soil, which is effective only across short distances or in porous soils [99]. Diffusive flows restricted only to the nest openings would not suffice to ventilate a serial nest, as demonstrated by the invariant $CO_2$ levels recorded when no wind was blowing over the surface of laboratory serial nests. Besides simple diffusion, another plausible ventilation mechanism is the presence of turbulent eddies at the tunnel entrances that create pulsatory changes in pressure inside tunnels, thus promoting mixing of air between the nest interior and the outside [100,101]. Nevertheless, our results indicate that this mechanism would not suffice to reduce $CO_2$ levels inside serial nests via the nest opening. Present results are in accordance with our previous measurements in field, multiserial *Atta* nests, which indicated that air circulation through superficial tunnels connected to the outside indeed occurred, but it did not directly renew the air inside the nest chambers [3]. As a result, $CO_2$ levels inside fungus chambers were similar to those of the nearby soil. Therefore, colony respiration completely depended on diffusive flows between the chamber air and the surrounding air phase of the soil [3]. Because of the lack of gas exchanges with the free atmosphere, i.e. the lack of wind-induced flows through the nest tunnels, a dynamic equilibrium between $CO_2$ in the soil and inside *Acromyrmex* serial nests is expected to occur in the long term.

Shallow nests, on the contrary, not only benefit by being excavated near the soil surface, i.e. by facing lower soil $CO_2$ levels and having shorter diffusion distances to the surface, but also by having more than one tunnel connecting a given chamber to the outside that enables passive, wind-induced ventilation. Our results indicated that besides the wind-induced ventilation acting on interconnected openings located at different heights, which is based on the Bernoulli principle and already shown for three different *Atta* species [3,4,9], turbulent eddies at interconnected nest entrances located at the same soil level suffice to ventilate the fungus garden. Eddies at the tunnel entrances are caused by small changes in speed of a non-steady wind flowing over the soil surface [100,101]. For thatched nests, laboratory results suggested that two ventilation mechanisms are involved. First, the presence of

wind-induced ventilation is based on the Bernoulli principle. When winds blow over a mound, air speeds up and moves faster at the top than at the base, resulting in a continuous gradient of decreasing pressure from the base to its top. Given the porous nature of the thatch in *Acromyrmex*, it results in mass movements of air through the thatch wall. The second mechanism is diffusion through the porous thatch structure that also provides a large surface for exchanges. It is important to point out that the innovative building behaviours exhibited by thatching *Acromyrmex* leaf-cutting ants resulted in a unique nest architecture able to promote a massive reduction of $CO_2$ levels inside fungus gardens through diffusive flows, as compared with the other two architectures in which diffusive flows alone through the nest openings were lacking.

In fact, field measurements aimed at comparing the two innovative architectural types showed that $CO_2$ concentrations inside *A. heyeri* thatched nests were on average lower than inside *A. lundii* shallow nests. In addition, they decreased much more strongly in *A. heyeri* nests than in *A. lundii* nests at the same wind speed. Although this difference can be assumed to depend on differences in nest architecture, the extent to which the lower concentrations inside thatched nests are a consequence of enhanced ventilation has to be cautiously addressed. Measured $CO_2$ concentrations depended on both the production by inhabiting organisms and the decrease directly caused by any ventilation mechanism involved. Given that during field measurements we were not able to assess fungus garden or colony sizes, differences in $CO_2$ inside nests could also arise because, for instance, *A. lundii* nests were larger than *A. heyeri* nests. Furthermore, it has to be considered that fungus gardens are located in a soil phase that acts as both a source and a sink of $CO_2$. Since soil levels of $CO_2$ are always many times higher than those of the atmosphere and increase with depth [2,102], levels inside serial and shallow nests are always expected to be higher than inside thatched nests. Nevertheless, we consider that our laboratory results, aimed at evaluating nest ventilation while controlling for potential effects of soil and fungus size, reinforce the idea that lower $CO_2$ levels inside thatched nests are the result of improved ventilation.

Our field experiment with airtight-sealed nests of *A. heyeri* indicated that thatched nests could be ventilated via mass movements of air driven by thermal convection, demonstrating the involvement of a third ventilation mechanism beyond diffusion and wind-induced flows as discussed above. Convective flows will occur as soon as the nest temperature inside the mound reaches values higher than the environment. Warm, less-dense air is expected to leave the nest through the upper part of the thatch, resulting in $CO_2$ emission into the atmosphere, followed by inflow of fresh air at its base because of the negative pressure generated inside. Thatched nests may achieve high nest temperatures because of their location above ground and their solar exposition. Together with the insulating properties of the nest material, they are expected to sustain thermal differences with the environment [46] that promote nest ventilation via thermal convection, as observed in termite nests [103,104]. In fact, long-term field measurements have shown that, in a yearly average, a fungus garden inside an *Acromyrmex* thatched nest achieves a temperature surplus of *ca* 5°C as compared with the environment [46]. Although there is no information about diel patterns of $CO_2$ concentrations inside thatched nests built by other *Acromyrmex* species, or its emissions of $CO_2$ at night, the relationship between daily temperature variation and $CO_2$ emission into the atmosphere resembles that of thatched nests of *Formica* wood ants in the Northern Hemisphere [105–109].

Taken together, our laboratory and field results in *Acromyrmex* support the idea that, in the long term, $CO_2$ levels inside the nest depend on the species-specific nest architectures, and that a particular nest architecture determines which passive mechanisms may operate to ventilate the nest. Nevertheless, $CO_2$ levels would not only depend on the degree of nest connectivity to the atmosphere, but also on the exchanges of $CO_2$ with the soil matrix. It is important to emphasize that the use of plastic chambers mimicking a totally impermeable soil during experiments certainly precluded any air movements through the chamber walls. In a field nest surrounded by a porous soil, however, wind-induced pulsatory changes in pressure at the nest entrance may potentially draw underground gases through the single nest opening, yet such flows would merely draw $CO_2$-rich air from the soil into the nest environment. Thus, inside serial nests, $CO_2$ concentration would tend to reach a dynamic equilibrium with that of the soil in the long term, as discussed above. If higher, $CO_2$ would diffuse into the soil, and vice versa, being such diffusive flows mainly dependent on the porosity of the surrounding soil. The main way to reach nest $CO_2$ levels lower than those of the surrounding soil would be to exchange air with the atmosphere via the nest tunnels, yet our results did not provide evidence for that in serial nests. However, air exchanges with the atmospheric air might occur, yet across the soil matrix. When the soil temperature is higher than that of the atmosphere, air will move upwards and leave the soil, leading to air renewal and to a decrease of $CO_2$ levels in the soil

superficial layers. Given that the magnitude of air exchanges between soil and atmosphere, driven by nocturnal thermal convection, decreases with increasing depth [2,102], those chambers of serial nests located near the soil surface will benefit by exchanging air with an air phase of the soil with lower $CO_2$ levels, as compared with deeper chambers [3]. It results in increased $CO_2$ levels inside chambers of serial nests, the deeper their location. Mass movements of air occurring when the outside air temperature is lower than inside the nest were already reported by Jacoby [110] and Fernandez-Bou *et al.* [111] in the serial *Atta sexdens* and *A. cephalotes* nests, respectively. Shallow *Acromyrmex* nests, which are located at the uppermost soil layers, would benefit even more than serial nests, because mass movements of nest air into the atmosphere, driven by thermal convection, are more pronounced at the superficial soil layers, as indicated above. In addition, the architectural innovation of having a low number of fungus chambers with multiple connections to the outside promotes nest ventilation by wind-induced flows. Finally, thatched nests are located at the interface between the soil and the atmosphere, and hence, they have a number of advantages as compared with serial and shallow nests. First, a thatched nest is much less influenced by gas exchanges with high $CO_2$ levels from the soil than the other nest types. Second, it ventilates directly to the atmosphere through the thatch material by diffusion, wind-induced ventilation and temperature-driven mass movements of air. In summary, $CO_2$ levels in ancestral serial nests are largely determined by the high $CO_2$ levels of the soil, lacking in addition a direct exchange with the atmospheric air via the nest tunnels, driven by wind-induced flows. Shallow nests have a significant, wind-driven exchange of air with the atmosphere via their nest tunnels, and the more innovative thatched nests exhibit the highest level of air exchanges with the atmosphere along with a reduced interaction with the high $CO_2$ levels of the soil.

The question arises, however, whether poorly ventilated nests, as the shallow and serial nests, are exposed to $CO_2$ levels harmful for the respiration of their fungus garden. All $CO_2$ values inside fungus gardens we have measured in the field fall in the range of concentrations selected by *Acromyrmex* workers for cultivating fungus and rearing brood, i.e. between 1 and 3% [5,6], being below the levels around 5% beyond which the respiration of the fungus garden is negatively affected in *Atta sexdens* [4]. We consider the values reported for the fungus of *A. sexdens* as indicative of average $CO_2$ levels that affect fungus respiration, since the *Atta sexdens* fungus belongs to the same clade as most *Acromyrmex* fungi [95]. Taking into account such levels, fungus garden respiration would not be compromised in those *Acromyrmex* species inhabiting shallow and thatched nests, i.e. those exhibiting the architectural innovations as compared with the ancestral nest type.

Answering the question of whether poorly ventilated serial nests are indeed exposed to $CO_2$ levels that negatively affect fungus respiration would require a consideration of the full range of $CO_2$ concentrations, from the soil surface to the deepest nest chamber. As discussed above, carbon dioxide levels in fungus gardens of serial nests should reach a dynamic equilibrium with those of the surrounding soil, which increase with depth. As we have previously measured in *Atta* nests, $CO_2$ concentrations in fungus chambers tend to increase linearly with the soil depth at which they are located, from 3 to 4.5% for depths of 1.5 to 3 m, respectively [3]. Such a trend of increasing $CO_2$ levels with soil depth has also been observed in colonies of *Pogonomyrmex* seed-harvester ants [112]. In fact, whether the soil acts as a sink or source for $CO_2$ does not only depend on soil depth, but also on soil temperature, moisture content and physical properties [2,102,113]. To date, our measurements on *Acromyrmex* fungus gardens inside shallow nests, superficially located between 0.2 and 0.5 m depth, averaged $CO_2$ concentrations around 2%. Several reports indicate that fungus gardens of serial *Acromyrmex* nests of different species can be located around 1.5–2 m depth [25,59], and even deeper [64,67]. Even for *Atta* nests, data on chamber depths and $CO_2$ concentrations inside fungus garden are incomplete. The deepest $CO_2$ measurements were obtained inside *Atta* fungus chambers located at 3 m depth [3], yet gardens can be located even at a depth of 7 m [21].

## 4.2. Selection pressures acting on nest architecture in leaf-cutting ants

Taken together, our current results and data we have previously published indicate that thatched nests, and shallow to a lesser extent, are not constrained by harmful levels of $CO_2$. Serial nests with the ancestral arrangement, on the contrary, may be constrained depending on the $CO_2$ levels in the soil phase and the depth of their chambers. Avoidance of unsuitable $CO_2$ levels inside *Acromyrmex* serial nests is therefore expected to rely more on fungus and brood relocation within the nest [5,6] than on ventilatory gas exchanges, as also expected for the serial nests of *Amoimyrmex* and *Atta*. We suggest that nest depth is the main feature directly affecting the ventilation of fungus chambers. Nest depth is in addition important for the control of nest humidity and temperature. Across its distribution range, there is a

relationship between soil temperatures and nesting habits in the genus *Acromyrmex*, from thatched nests in those species inhabiting temperate soils to serial nests in those species inhabiting tropical soils. In addition, several *Acromyrmex* species are known to change nesting habits between thatched and shallow nests depending on soil temperature, i.e. building a thatch when inhabiting cold soils, and excavating shallow nests in warmer soils [31]. Such a phenotypic plasticity in nest architecture appears to be associated with short-term responses to environmental variables. *Acromyrmex* workers are known to modify the nest structure in the short term as a response to changes in temperature and humidity levels unsuitable for fungus and brood [92,93,114], and to select suitable soil temperatures for nest excavation [31], which leads to a temperature-dependent selection of nest depth. Besides, workers trade off regulation of one climatic variable for others by modifying the architecture of the nest. *A. heyeri* workers, for instance, close nest entrances and openings to maintain high humidity inside fungus gardens, at the expense of temperature regulation and air circulation [92,114], since fungus gardens rapidly desiccate when exposed to dry conditions.

In addition, *Acromyrmex* workers also use $CO_2$ as an orientation cue during digging and show marked preferences for $CO_2$ levels for excavation. When digging, workers avoided levels of 4% and preferred places with levels of 1% $CO_2$ [7]. Given that $CO_2$ concentration increases with depth, should this preference imply that the nest volume would tend to grow near the soil surface because of the avoidance of deep soil layers with higher $CO_2$ levels? Without disregarding the importance of $CO_2$-sensitive digging, we consider that the architectural innovations advantageous for nest ventilation, i.e. the building of shallow and thatched nests, evolved under selection pressures acting mainly on temperature regulation. The fact that most leaf-cutting ant species, i.e. the complete genus *Atta*, *Amoimyrmex* and several *Acromyrmex* species, inhabit serial nests reinforces this view. Although a serial arrangement of nest chambers constrains nest ventilation, it provides benefits in terms of humidity and temperature regulation, because soil moisture increases with depth [2], and soil temperatures suitable for fungus rearing are found at the deep layers of tropical soils instead of at the surface level [31].

The current hypothesis about the evolution of leaf-cutting ants locates their centre of origin in the South or Central American tropics, from where they expanded into north and southern South America [12–14,95]. When considering the relationship between evolution of leaf-cutting ants and their nest architectures, although the ancestral serial nest type in the sister group *Trachymyrmex* may preclude the occurrence of ventilatory gas exchanges with the atmosphere via their tunnels, their small colonies are expected to be located at low soil depths and therefore exposed to non-detrimental, low $CO_2$ levels. From there, leaf-cutting ant genera followed different evolutionary patterns. While *Amoimyrmex* and *Atta* retained a serial nest architecture that constrains ventilatory gas exchanges, several *Acromyrmex* evolved two nesting types, shallow and thatched architectures that resulted in improved nest ventilation. The *Atta* genus is more diverse in the tropical regions of South America, while the genus *Acromyrmex* reaches its highest species diversity in the subtropical southern South America [115,116]. In these regions, moderate, sub-optimal temperatures as occurring in temperate soils strongly selected for the habit of nesting near or above the soil surface level [31,46,117], i.e. leading to the evolution of architectural innovations such as shallow and thatched nests. We suggest that such architectural innovations in *Acromyrmex* evolved under selective forces related to temperature control, with a concomitant benefit, as demonstrated in the present study, in terms of improved gas exchanges and avoidance of the unsuitable $CO_2$ levels that occur at deep soil layers.

Data accessibility. The datasets supporting this article have been uploaded as part of the electronic supplementary material.

The data are provided in the electronic supplementary material [118].

Authors' contributions. Conceptualization: M.B., F.R. and D.R. Funding acquisition: F.R., M.B. and D.R. Data acquisition and formal analysis: M.B. Writing first draft: M.B. Writing review and editing: M.B., F.R. and D.R. Approved the version to be published: M.B., F.R. and D.R. Agreement to be accountable for all aspects of the work in ensuring that questions related to the accuracy or integrity of any part of the work are appropriately investigated and resolved: M.B., D.R. and F.R.

Competing interests. Authors do not have competing interests.

Funding. This work was partially supported by the German Research Foundation (DFG, grant no. SFB 554/TP E1), the Agencia Nacional de Investigación e Innovación (ANII Uruguay, grant Fondo Maria Viñas FMV 156057 to M.B.), the Comisión Sectorial de Investigación Científica (CSIC-UDELAR Uruguay, grant no. I+D 489 to M.B.) and the Department of Behavioural Physiology and Sociobiology, Biocenter, University of Würzburg, Germany, headed by Prof. Dr Wolfgang Rössler. D.R. was supported by the Postdoc Plus funding program of the Graduate School of

Life Sciences (GSLS), University of Würzburg, Germany and a Postdoctoral Fellowship of the Agencia Nacional de Investigación e Innovación (ANII, grant no. PD_NAC_2015_1_108641), Uruguay.

Acknowledgements. M.B. thanks Leticia Tejera for her invaluable support. M.B. thanks Christian Rabeling for his comments on the first idea of this work. Thanks are also due to two anonymous reviewers for their comments that helped to improve the manuscript.

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
