## [Peer Review File · Royal Society Open Science]

Review History

RSOS-210907.R0 (Original submission)

Review form: Reviewer 1

Is the manuscript scientifically sound in its present form?

No

Are the interpretations and conclusions justified by the results?

No

Is the language acceptable?

Yes

Do you have any ethical concerns with this paper?

No

Have you any concerns about statistical analyses in this paper?

No

Recommendation?

Accept with minor revision (please list in comments)

Comments to the Author(s)

This paper presents and addresses interesting questions about the functional role of nest architecture in respiration of fungus cultivating ant colonies. The study builds on previous work on the biological system by the authors. The combination of simplified lab experiments and field measurements casts light on the selective pressures driving the development of architectural motifs identified in the paper. Most of the results are far from surprising, but are worthwhile and relevant to the motivated questions. The overall narrative and discussion is clear and engaging.

There were few points of confusion for me that would be good to resolve.

Regarding the lab experiments, some of the underlying logic was not obvious:

1. There was meaningful discussion of diffusive mass transport between nest and surrounding soil, particularly with the serial nests. How those dynamics compare with other possible modes of exchange seemed to be the key question, and wind-driven exchange is counted among those others. How wind should be expected to cause exchange is not obvious, but it seems plausible that either transients in the flow near the entrance (like that cited in Ref. 61, but with a single entrance), or create pulsatory pressure differentials between it and internal surfaces, or that steady flow creates a constant pressure differential between the same. In either case, the amount of exchange for given wind speed would depend on the permeability of the soil. It seems intuitive that the permeability would be small enough to make this effect negligible (at least for reasonable wind values), but the fact that our intuition is not great for such seeping flows and complicated geometries would be reason to do the experiment. In the lab experiment, perfectly impermeable containers were used for the fungus gardens as well as peduncle, and there is no sign of exchange. The result seems to demonstrate the assumption about soil permeability more than it rules out a particular hypothesis about wind. (I would wonder how much total resistance I would experience upon forming a tight seal with the entrance and trying to blow air in...) In any case, permeability of the soil with respect to pressure-driven flow (due to thermal gradients) is referenced directly in the text (ie. page 10, line 20). Perhaps the use of impermeable containers could be justified and the question addressed by the experiment could be more explicitly stated.
2. Relatedly: It wasn't clear to me the role of live fungus in testing the premise of the experiment. A simpler result (that no wind-driven exchange was occurring) could be more easily achieved with an initial volume or controlled input of CO₂ in one chamber, monitored over time with with/without wind tangential to the opening. The additional complexity introduced with active production of CO₂ via live fungus, again, seems to be motivated by the need to rule out a competing hypothesis that wasn't apparent to me. Also, why not report the CO₂ values over time, rather than take the one snapshot at one hour? It might be satisfying/revealing to see 4 independent curves of increasing CO₂ as the fungus respire.

Regarding field measurements:

1. Fig. 4C was a little mystifying to me. The correlation between wind and CO₂ levels, as presented, appears way weaker than I would have thought, and I wonder if there was some structure in the data that was not properly accounted for when processing the data. Particularly the many points showing positive correlations between wind and CO₂ are hard to accept. How should we understand them? Over what interval in time were the wind speed and CO₂ measurements grouped? I can imagine that a wind event following some stagnation could churn up high CO₂ air from the nest toward the sensor, but the very same event should lower the CO₂ over a somewhat longer interval. Could that type of situation be reflected in the results as reported? Perhaps a different method of processing the raw data (eg. low pass filter) would extract the physically meaningful signal from high frequency noise. If another method doesn't

work, it could be useful for an interested reader to see more of the raw data, not unlike that shown in Fig. 4B, in the SI.

A general issue that kept coming to mind while reading is the role of the harder-to-know rate and spatial density of CO₂ production in putting pressure on the different exchange mechanisms. In some places, "CO₂ levels" seem to take the place of the more relevant notion of rates of exchange. For instance, that "... levels of CO₂ inside fungus gardens in serial nests would equilibrate with those of the soil phase, and vary depending on the depth..." assumes production of CO₂ is very slow compared to the rate of diffusion into the soil. Comment on the validity of this assumption is warranted. Minor point: Rather than truly equilibrate, I'd say the long-term CO₂ concentration profile would reflect a 'steady-state' condition or 'dynamic equilibrium'.

In any case, it makes sense that increased size of the nest, and therefore increased rate of production of CO₂ plays a dominant role in requiring adaptation in ventilation mechanisms. Of the large serial nests (like *Atta*) without wind-driven exchange, according to the results, has there been analysis of the size and spacing of chambers that balance the CO₂ production rate with the diffusive exchange rate in the soil? Just a thought.

Review form: Reviewer 2

Is the manuscript scientifically sound in its present form?

Yes

Are the interpretations and conclusions justified by the results?

Yes

Is the language acceptable?

Yes

Do you have any ethical concerns with this paper?

No

Have you any concerns about statistical analyses in this paper?

No

Recommendation?

Accept with minor revision (please list in comments)

Comments to the Author(s)

The authors perform field and laboratory assays to determine how three different nest architectural plans influence CO₂ ventilation in *Acromyrmex* fungus-farming ant colonies. CO₂ levels within nests were shown to be strongly dependent on nest architecture, with thatched mounds acting as superior means of increasing gas exchange. This paper will be of broad interest to readers interested in functional morphology of the "extended organism" and the evolution of unique blueprints in response to simple tradeoffs between the regulation of humidity, temperature and ventilation. The interpretation of the results is well supported by the data, and I found the figures especially appealing (figure 3, in particular). I have several suggestions that would improve this draft of the manuscript:

1. You note that serial chambers architecture is basal/more ancient, but do not mention the lower attines that build alternative nest types, which are not serial. For instance, the yeast growing *Cyphomyrmex*, and some *Apterostigma* spp. maintain shallow fungus gardens under

loose leaf litter, in cavities in wood, and on in curled leaves, with a veil of fungus that acts as a “skin” over the colony and exposed garden. I agree that “ancestral ants” likely produced a simple serial nest. However, you are underrepresenting the diversity in the lower attines.

2. You state that “To our knowledge there are no studies on evolutionary trade-offs that may account for differences in architectural traits in ant nests.” However, I would argue that you and your colleagues have already demonstrated that turrets are an adaptive feature of nest architecture, in the contexts of ventilation and flooding. Several other studies of the adaptive function of chamber size and arrangement have been undertaken (Tschinkel 2018, *Solenopsis invicta*; Vaes et al. 2020, *Myrmica ruba* – though this study has design issues). Also, the function of nest mound architecture in preventing inundation by water, and funnel-entrances and narrow entrance shafts preventing invasion by army ants in *Ectatomma* and other ant species. In other words, I would caution you not to suggest this is the first attempt to understand evolutionary tradeoffs of nest architecture in ants.

3. Introduction page 1, Line 27 (opening sentence): Change “could” to “can”

4. You provide estimates of *Atta* chamber number, but roughly how many chambers or garden chambers do serial *Acromyrmex* colonies have in nature?

5. Page 4, line 46: change “selected for to” to “selected to”

6. I am glad that you made your comparisons “between the CO₂ concentration for each measurement at a given wind speed minus the average concentration measured in total absence of wind for the nest under consideration.” However, your laboratory design prevents you from comparing between nest designs, because total chamber volume differs considerably between treatments. You may want to explain why you did not standardize chamber volume and make a note on figure 3 to indicate that comparisons cannot be made between nest designs.

7. Add sample sizes to the caption of fig 3.,

8. Discussion: The following sentence about tradeoffs between nest types is key to your argument, and should be emphasized earlier in your discussion, and perhaps even the introduction: “Although a serial arrangement of nest chambers constrains nest ventilation, it provides benefits in terms of humidity and temperature regulation, because soil humidity increases with depth [59], and suitable soil temperatures for fungus rearing are found in the deep layers of tropical soils instead of at the surface level [29].” Unless you very clearly describe these tradeoffs early on, readers will question your assertion about adaptive architecture, given that *Atta* colonies achieve great success with a serial nest design.

9. The discussion could be streamlined and shortened.

10. I enjoyed the supplementary videos.

Decision letter (RSOS-210907.R0)

Dear Dr Bollazzi

The Editors assigned to your paper RSOS-210907 "CO₂ levels and ventilation in *Acromyrmex* nests: significance and evolution of architectural innovations in leaf-cutting ants" have now received comments from reviewers and would like you to revise the paper in accordance with the reviewer comments and any comments from the Editors. Please note this decision does not guarantee eventual acceptance.

Please submit your revised manuscript and required files (see below) no later than 21 days from today's (ie 08-Jul-2021) date. Note: the ScholarOne system will 'lock' if submission of the revision is attempted 21 or more days after the deadline. If you do not think you will be able to meet this deadline please contact the editorial office immediately.

on behalf of Prof Kevin Padian (Subject Editor)
openscience@royalsociety.org

Associate Editor Comments to Author:

Please carefully respond to the comments below - it is unlikely that further rounds of revision will be possible.

Reviewer comments to Author:

Reviewer: 1

Comments to the Author(s)

This paper presents and addresses interesting questions about the functional role of nest architecture in respiration of fungus cultivating ant colonies. The study builds on previous work on the biological system by the authors. The combination of simplified lab experiments and field measurements casts light on the selective pressures driving the development of architectural motifs identified in the paper. Most of the results are far from surprising, but are worthwhile and relevant to the motivated questions. The overall narrative and discussion is clear and engaging.

There were few points of confusion for me that would be good to resolve.

Regarding the lab experiments, some of the underlying logic was not obvious:

1. There was meaningful discussion of diffusive mass transport between nest and surrounding soil, particularly with the serial nests. How those dynamics compare with other possible modes of exchange seemed to be the key question, and wind-driven exchange is counted among those others. How wind should be expected to cause exchange is not obvious, but it seems plausible that either transients in the flow near the entrance (like that cited in Ref. 61, but with a single entrance), or create pulsatory pressure differentials between it and internal surfaces, or that steady flow creates a constant pressure differential between the same. In either case, the amount of exchange for given wind speed would depend on the permeability of the soil. It seems intuitive that the permeability would be small enough to make this effect negligible (at least for reasonable wind values), but the fact that our intuition is not great for such seeping flows and complicated geometries would be reason to do the experiment. In the lab experiment, perfectly impermeable containers were used for the fungus gardens as well as peduncle, and there is no sign of exchange. The result seems to demonstrate the assumption about soil permeability more than it rules out a particular hypothesis about wind. (I would wonder how much total resistance I would experience upon forming a tight seal with the entrance and trying to blow air in...) In any case, permeability of the soil with respect to pressure-driven flow (due to thermal gradients) is referenced directly in the text (ie. page 10, line 20). Perhaps the use of impermeable containers could be justified and the question addressed by the experiment could be more explicitly stated.
2. Relatedly: It wasn't clear to me the role of live fungus in testing the premise of the experiment. A simpler result (that no wind-driven exchange was occurring) could be more easily achieved with an initial volume or controlled input of CO₂ in one chamber, monitored over time with with/without wind tangential to the opening. The additional complexity introduced with active production of CO₂ via live fungus, again, seems to be motivated by the need to rule out a competing hypothesis that wasn't apparent to me. Also, why not report the CO₂ values over time, rather than take the one snapshot at one hour? It might be satisfying/revealing to see 4 independent curves of increasing CO₂ as the fungus respire.

Regarding field measurements:

1. Fig. 4C was a little mystifying to me. The correlation between wind and CO₂ levels, as presented, appears way weaker than I would have thought, and I wonder if there was some structure in the data that was not properly accounted for when processing the data. Particularly the many points showing positive correlations between wind and CO₂ are hard to accept. How should we understand them? Over what interval in time were the wind speed and CO₂ measurements grouped? I can imagine that a wind event following some stagnation could churn up high CO₂ air from the nest toward the sensor, but the very same event should lower the CO₂ over a somewhat longer interval. Could that type of situation be reflected in the results as reported? Perhaps a different method of processing the raw data (eg. low pass filter) would extract the physically meaningful signal from high frequency noise. If another method doesn't work, it could be useful for an interested reader to see more of the raw data, not unlike that shown in Fig. 4B, in the SI.

A general issue that kept coming to mind while reading is the role of the harder-to-know rate and spatial density of CO₂ production in putting pressure on the different exchange mechanisms. In some places, "CO₂ levels" seem to take the place of the more relevant notion of rates of exchange. For instance, that "... levels of CO₂ inside fungus gardens in serial nests would equilibrate with those of the soil phase, and vary depending on the depth..." assumes production of CO₂ is very slow compared to the rate of diffusion into the soil. Comment on the validity of this assumption is warranted. Minor point: Rather than truly equilibrate, I'd say the long-term CO₂ concentration profile would reflect a 'steady-state' condition or 'dynamic equilibrium'.

In any case, it makes sense that increased size of the nest, and therefore increased rate of production of CO₂ plays a dominant role in requiring adaptation in ventilation mechanisms. Of the large serial nests (like *Atta*) without wind-driven exchange, according to the results, has there been analysis of the size and spacing of chambers that balance the CO₂ production rate with the diffusive exchange rate in the soil? Just a thought.

Reviewer: 2

Comments to the Author(s)

The authors perform field and laboratory assays to determine how three different nest architectural plans influence CO₂ ventilation in *Acromyrmex* fungus-farming ant colonies. CO₂ levels within nests were shown to be strongly dependent on nest architecture, with thatched mounds acting as superior means of increasing gas exchange. This paper will be of broad interest to readers interested in functional morphology of the “extended organism” and the evolution of unique blueprints in response to simple tradeoffs between the regulation of humidity, temperature and ventilation. The interpretation of the results is well supported by the data, and I found the figures especially appealing (figure 3, in particular). I have several suggestions that would improve this draft of the manuscript:

1. You note that serial chambers architecture is basal/more ancient, but do not mention the lower attines that build alternative nest types, which are not serial. For instance, the yeast growing *Cyphomyrmex*, and some *Apterostigma* spp. maintain shallow fungus gardens under loose leaf litter, in cavities in wood, and on in curled leaves, with a veil of fungus that acts as a “skin” over the colony and exposed garden. I agree that “ancestral ants” likely produced a simple serial nest. However, you are underrepresenting the diversity in the lower attines.

2. You state that “To our knowledge there are no studies on evolutionary trade-offs that may account for differences in architectural traits in ant nests.” However, I would argue that you and your colleagues have already demonstrated that turrets are an adaptive feature of nest architecture, in the contexts of ventilation and flooding. Several other studies of the adaptive function of chamber size and arrangement have been undertaken (Tschinkel 2018, *Solenopsis invicta*; Vaes et al. 2020, *Myrmica ruba* – though this study has design issues). Also, the function of nest mound architecture in preventing inundation by water, and funnel-entrances and narrow entrance shafts preventing invasion by army ants in *Ectatomma* and other ant species. In other words, I would caution you not to suggest this is the first attempt to understand evolutionary tradeoffs of nest architecture in ants.

3. Introduction page 1, Line 27 (opening sentence): Change “could” to “can”

4. You provide estimates of *Atta* chamber number, but roughly how many chambers or garden chambers do serial *Acromyrmex* colonies have in nature?

5. Page 4, line 46: change “selected for to” to “selected to”

6. I am glad that you made your comparisons “between the CO₂ concentration for each measurement at a given wind speed minus the average concentration measured in total absence of wind for the nest under consideration.” However, your laboratory design prevents you from comparing between nest designs, because total chamber volume differs considerably between treatments. You may want to explain why you did not standardize chamber volume and make a note on figure 3 to indicate that comparisons cannot be made between nest designs.

7. Add sample sizes to the caption of fig 3.,

8. Discussion: The following sentence about tradeoffs between nest types is key to your argument, and should be emphasized earlier in your discussion, and perhaps even the introduction: "Although a serial arrangement of nest chambers constrains nest ventilation, it provides benefits in terms of humidity and temperature regulation, because soil humidity increases with depth [59], and suitable soil temperatures for fungus rearing are found in the deep layers of tropical soils instead of at the surface level [29]." Unless you very clearly describe these tradeoffs early on, readers will question your assertion about adaptive architecture, given that *Atta* colonies achieve great success with a serial nest design.

9. The discussion could be streamlined and shortened.

10. I enjoyed the supplementary videos.

===PREPARING YOUR MANUSCRIPT===

===PREPARING YOUR REVISION IN SCHOLARONE===

Author's Response to Decision Letter for (RSOS-210907.R0)

See Appendix A.

RSOS-210907.R1 (Revision)

Review form: Reviewer 1

Is the manuscript scientifically sound in its present form?

No

Are the interpretations and conclusions justified by the results?

No

Is the language acceptable?

Yes

Do you have any ethical concerns with this paper?

No

Have you any concerns about statistical analyses in this paper?

No

Recommendation?

Accept with minor revision (please list in comments)

Comments to the Author(s)

I appreciate the authors' work in revising the manuscript. I do have some remaining concerns that I hope can be addressed without major changes:

1. The role of soil porosity in wind-driven ventilation. This may be addressed in words without much additional work, but will serve to justify the wind experiments. I apologize if I had not effectively expressed my question before. If you suck on a straw stuck deep in the sand, you can pull some air out, because the permeability of the sand integrated over a large area underground gives some less-than-infinite resistance. If the straw ends in an impermeable chamber, you can't drive flow. This is why I had insisted that the potential effect of the wind in driving some ventilation necessarily depends on the porosity of the soil, in principle. If it is hard to force air into or out of the entrance holes of the nests in the field, it would indicate my point is probably not practically relevant. I wouldn't know which to guess, but it changes the significance of conclusions drawn from the lab experiments.

2. The direction of net flux of CO₂ underground. Between my first reading of the manuscript and the revision, my confusion has shifted. What I would like to visualize better is a physically consistent picture of how CO₂ gets from source to atmosphere. As noted in the manuscript, the soil itself is producing CO₂, and due to this production, the CO₂ levels are known to increase with depth -- and this is important for the issue of transport between nest and soil. However, the more specific nature of this balance is unclear. I want to be careful about my terms here:
Equilibrium = no concentration difference between 2 bodies, therefore no transport between them (via Fick's law)

Steady state = condition in which the net flux of CO₂ out of any element is equal to its rate of production, such that the level measured at any location is constant over time (it isn't accumulating anywhere at steady state)

I guess "dynamic equilibrium" is a valid equivalent to "steady state" above.

The added lines 427-443 confused me here. The suggestion that CO₂ diffuses into the nest at larger depths leads to a couple different conclusions which seem important and counterintuitive.

(At first reading, one might interpret the nest metabolism runs backwards to accommodate the boundary conditions...) The direction of flux between nest and soil should be determined by the local rate of CO₂ production, not the CO₂ levels in soil without the nest. Do we assume that the soil at deeper depths produces more CO₂ volumetrically, or are the observed values representative of the steady state of constant flux toward the surface of CO₂ produced at constant rate at different levels? If the latter, for instance, CO₂ will still flow from nest to soil (because the production rate is volumetrically greater in the nest), but the steady state concentration of the nest will simply surpass the ambient value at that depth. The former is really hard to imagine, but would necessarily mean that for every bit of flux into the nest from the neighboring soil (laterally), there must be that much more flux outward (presumably toward the surface) to be at steady state. Further comment and clarity here would be very helpful. (Analysis of a similar situation in Ref. 62 (Wilson, Kilgore 1978) was useful for me.)

Minor point:

"We agree that curves representing the change in the CO₂ values over time would be informative about the respiration physiology of the fungus and ants under the increasing CO₂ concentrations, yet our goal was not to open a discussion about this topic in the present work."

--The discussion which is open here does not just concern respiration physiology. How the CO₂ levels approach the reported values over time is relevant to a reader interested in the mass transfer dynamics (arguably the basic substance of this work). If the data is not available for this paper, that is ok, but I don't agree with the rationale for its omission.

Review form: Reviewer 2

Is the manuscript scientifically sound in its present form?

Yes

Are the interpretations and conclusions justified by the results?

Yes

Is the language acceptable?

Yes

Do you have any ethical concerns with this paper?

No

Have you any concerns about statistical analyses in this paper?

No

Recommendation?

Accept as is

Comments to the Author(s)

The authors have significantly remodeled the manuscript in response to both reviewers - including clarification of background, additional details of the methods and justification of the questions asked and hypotheses tested. The paper is ready for publication in my opinion.

Decision letter (RSOS-210907.R1)

Dear Dr Bollazzi

On behalf of the Editors, we are pleased to inform you that your Manuscript RSOS-210907.R1 "Carbon dioxide levels and ventilation in Acromyrmex nests: significance and evolution of architectural innovations in leaf-cutting ants" has been accepted for publication in Royal Society Open Science subject to minor revision in accordance with the referees' reports. Please find the referees' comments along with any feedback from the Editors below my signature.

Please submit your revised manuscript and required files (see below) no later than 7 days from today's (ie 20-Sep-2021) date. Note: the ScholarOne system will 'lock' if submission of the revision is attempted 7 or more days after the deadline. If you do not think you will be able to meet this deadline please contact the editorial office immediately.

on behalf of Kevin Padian (Subject Editor)
openscience@royalsociety.org

Associate Editor Comments to Author:
Comments to the Author:

Please see the remaining comments from Reviewer 1. While a further review process is unlikely, this is also contingent on your satisfying the Editors that you've taken the reviewer's concerns seriously and addressed them not only in the response to reviewers but the manuscript itself (please ensure a tracked-changes version of the - hopefully - final iteration of the paper is included when you resubmit). The reviewer also poses a question about the availability of data: in the event you have data regarding this point, please ensure it is made accessible in-line with the journal's open data policy (and check with the editorial office if anything is unclear here); if

you do not have direct access to data, but can point to a supporting reference or two, this would be helpful.

Reviewer comments to Author:

Reviewer: 1

Comments to the Author(s)

I appreciate the authors' work in revising the manuscript. I do have some remaining concerns that I hope can be addressed without major changes:

1. The role of soil porosity in wind-driven ventilation. This may be addressed in words without much additional work, but will serve to justify the wind experiments. I apologize if I had not effectively expressed my question before. If you suck on a straw stuck deep in the sand, you can pull some air out, because the permeability of the sand integrated over a large area underground gives some less-than-infinite resistance. If the straw ends in an impermeable chamber, you can't drive flow. This is why I had insisted that the potential effect of the wind in driving some ventilation necessarily depends on the porosity of the soil, in principle. If it is hard to force air into or out of the entrance holes of the nests in the field, it would indicate my point is probably not practically relevant. I wouldn't know which to guess, but it changes the significance of conclusions drawn from the lab experiments.

2. The direction of net flux of CO₂ underground. Between my first reading of the manuscript and the revision, my confusion has shifted. What I would like to visualize better is a physically consistent picture of how CO₂ gets from source to atmosphere. As noted in the manuscript, the soil itself is producing CO₂, and due to this production, the CO₂ levels are known to increase with depth -- and this is important for the issue of transport between nest and soil. However, the more specific nature of this balance is unclear. I want to be careful about my terms here:
Equilibrium = no concentration difference between 2 bodies, therefore no transport between them (via Fick's law)

Steady state = condition in which the net flux of CO₂ out of any element is equal to its rate of production, such that the level measured at any location is constant over time (it isn't accumulating anywhere at steady state)

I guess "dynamic equilibrium" is a valid equivalent to "steady state" above.

The added lines 427-443 confused me here. The suggestion that CO₂ diffuses into the nest at larger depths leads to a couple different conclusions which seem important and counterintuitive. (At first reading, one might interpret the nest metabolism runs backwards to accommodate the boundary conditions...) The direction of flux between nest and soil should be determined by the local rate of CO₂ production, not the CO₂ levels in soil without the nest. Do we assume that the soil at deeper depths produces more CO₂ volumetrically, or are the observed values representative of the steady state of constant flux toward the surface of CO₂ produced at constant rate at different levels? If the latter, for instance, CO₂ will still flow from nest to soil (because the production rate is volumetrically greater in the nest), but the steady state concentration of the nest will simply surpass the ambient value at that depth. The former is really hard to imagine, but would necessarily mean that for every bit of flux into the nest from the neighboring soil (laterally), there must be that much more flux outward (presumably toward the surface) to be at steady state. Further comment and clarity here would be very helpful. (Analysis of a similar situation in Ref. 62 (Wilson, Kilgore 1978) was useful for me.)

Minor point:

"We agree that curves representing the change in the CO₂ values over time would be informative about the respiration physiology of the fungus and ants under the increasing CO₂ concentrations, yet our goal was not to open a discussion about this topic in the present work."

--The discussion which is open here does not just concern respiration physiology. How the CO₂ levels approach the reported values over time is relevant to a reader interested in the mass transfer dynamics (arguably the basic substance of this work). If the data is not available for this paper, that is ok, but I don't agree with the rationale for its omission.

Reviewer: 2

Comments to the Author(s)

The authors have significantly remodeled the manuscript in response to both reviewers - including clarification of background, additional details of the methods and justification of the questions asked and hypotheses tested. The paper is ready for publication in my opinion.

===PREPARING YOUR MANUSCRIPT===

===PREPARING YOUR REVISION IN SCHOLARONE===

Author's Response to Decision Letter for (RSOS-210907.R1)

See Appendix B.

Decision letter (RSOS-210907.R2)

Dear Dr Bollazzi,

I am pleased to inform you that your manuscript entitled "Carbon dioxide levels and ventilation in Acromyrmex nests: significance and evolution of architectural innovations in leaf-cutting ants" is now accepted for publication in Royal Society Open Science.

Our payments team will be in touch shortly if you are required to pay a fee for the publication of the paper (if you have any queries regarding fees, please see https://royalsocietypublishing.org/rsos/charges or contact authorfees@royalsociety.org).

on behalf of Professor Kevin Padian (Subject Editor)
openscience@royalsociety.org

Follow Royal Society Publishing on Twitter: @RSocPublishing
Follow Royal Society Publishing on Facebook:
<https://www.facebook.com/RoyalSocietyPublishing.FanPage/>

Read Royal Society Publishing's blog:
<https://royalsociety.org/blog/blogsearchpage/?category=Publishing>

Appendix A

Editorial Office of Royal Society Open Science

Montevideo, Uruguay / Würzburg, Germany, 14 August 2021

Dear Editors of Royal Society Open Science!

We would like to submit the REVISED version of our manuscript (RSOS-210907) entitled **“Carbon dioxide levels and ventilation in *Acromyrmex* nests: significance and evolution of architectural innovations in leaf-cutting ants”**, by Martin Bollazzi, Daniela Römer and Flavio Rocas, to be considered for publication in *Royal Society Open Science*.

We are very much indebted to the Reviewers, whose comments were very helpful to improve the revised version of our manuscript. We are providing a thoroughly revised version in which we addressed all the Reviewers' queries. We have now included a proper discussion about diffusive flows between fungus and soil, answered why we have used impermeable containers and a fungus garden instead of a CO₂ bolus during experiments, as well as improved Fig 4 as requested by reviewer 1. Besides, and based on the suggestions of reviewer 2, we have better introduced and discussed the architecture of basal fungus growing ants, the evolutionary trade-offs related to control of nest climate, as well as other minor concerns raised during revision.

All replies to the Reviewers' questions can be found in the 'response to reviewers' word file we uploaded to the Royal Society editorial manager.

We hope you will agree with our detailed answers and the corresponding changes in the text, when giving our manuscript further consideration.

Thank you again for considering our manuscript for publication in Royal Society Open Science!

sincerely yours,

Dr. Martin Bollazzi, Dra. Daniela Römer and Dr. Flavio Rocas

Reviewer comments to Author:

Reviewer: 1

Comments to the Author(s)

This paper presents and addresses interesting questions about the functional role of nest architecture in respiration of fungus cultivating ant colonies. The study builds on previous work on the biological system by the authors. The combination of simplified lab experiments and field measurements casts light on the selective pressures driving the development of architectural motifs identified in the paper. Most of the results are far from surprising, but are worthwhile and relevant to the motivated questions. The overall narrative and discussion is clear and engaging.

There were few points of confusion for me that would be good to resolve.

Reviewer 1 _ comment 1

Regarding the lab experiments, some of the underlying logic was not obvious:

1.

There was meaningful discussion of diffusive mass transport between nest and surrounding soil, particularly with the serial nests. How those dynamics compare with other possible modes of exchange seemed to be the key question, and wind-driven exchange is counted among those others.

How wind should be expected to cause exchange is not obvious, but it seems plausible that either transients in the flow near the entrance (like that cited in Ref. 61, but with a single entrance), or create pulsatory pressure differentials between it and internal surfaces, or that steady flow creates a constant pressure differential between the same.

In either case, the amount of exchange for given wind speed would depend on the permeability of the soil. It seems intuitive that the permeability would be small enough to make this effect negligible (at least for reasonable wind values), but the fact that our intuition is not great for such seeping flows and complicated geometries would be reason to do the experiment.

In the lab experiment, perfectly impermeable containers were used for the fungus gardens as well as peduncle, and there is no sign of exchange. The result seems to demonstrate the assumption about soil permeability more than it rules out a particular hypothesis about wind. (I would wonder how much total resistance I would experience upon forming a tight seal with the entrance and trying to blow air in...)

In any case, permeability of the soil with respect to pressure-driven flow (due to thermal gradients) is referenced directly in the text (ie. page 10, line 20). Perhaps the use of impermeable containers could be justified and the question addressed by the experiment could be more explicitly stated.

ANSWER Reviewer 1 _ comment 1:

DONE. We agree with the Reviewer that some of the underlying logic of the laboratory experiments was not obvious. We have therefore modified the Introduction and the Discussion sections in order to properly address these comments. We have explicitly stated that our goal was not to experimentally address the exchanges of CO₂ between the fungus and the soil phase. This is why we decided to experimentally control for potential diffusive flows through the soil structure, which depend on several variables such as soil porosity, soil moisture, etc., by using impermeable plastic containers that precluded them completely. Therefore, the role of other passive drivers of nest ventilation, such as wind-induced flows or convective flows, could be experimentally addressed, both in the laboratory and in the field. In the revised version, we have proceeded as follows to clarify our aims. First, we recognized that as a general rule, the fungus gardens exchange CO₂ with the soil via diffusive flows (Lines 43-51 / 62-65 in the Introduction, and 416-418 / 426-443 in the Discussion). Second, we argued the direction of such exchange may depend on the soil depth at which it occurs, i.e., from the fungus into the soil in shallow nests, and from the soil into deeper-placed chambers, since soil CO₂ levels increase with depth (Lines 43-51 in the Introduction, and 426-443 / 445-450 in the Discussion). Third, we clarified that our investigations were aimed at evaluating the specific influence of the different nest architectures on nest ventilation, in addition to the diffusive gas exchanges that are expected to occur between the nest and the soil phase, and the nest and the outside air (Lines 62-65 / 115-117 in the Introduction, 194-201 in the Methods, and 414-418 in the Discussions). This was achieved in the laboratory by controlling for other variables such as soil characteristics and nest temperature, as mentioned above (Lines 226-230 in the Methods).

Reviewer 1 _ comment 2

2. Relatedly: It wasn't clear to me the role of live fungus in testing the premise of the experiment. A simpler result (that no wind-driven exchange was occurring) could be more easily achieved with an initial volume or controlled input of CO₂ in one chamber, monitored over time with with/without wind tangential to the opening. The additional complexity introduced with active production of CO₂ via live fungus, again, seems to be motivated by the need to rule out a competing hypothesis that wasn't apparent to me. Also, why not report the CO₂ values over time, rather than take the one snapshot at one hour? It might be satisfying/revealing to see 4 independent curves of increasing CO₂ as the fungus respire.

ANSWER Reviewer 1 _ comment 2:

DONE: We agree with the Reviewer that evidence for nest ventilation could be obtained using different experimental designs and approaches. Among them, the use of a controlled input of CO₂, or of a CO₂ bolus of fixed volume, and the monitoring of the CO₂ concentration over time inside chambers, with or without tangential wind blowing above the nest openings. In the first method, the production rate of CO₂ of an ant nest would be simulated by the controlled input. In the second one, no CO₂ production would occur and the clearance time should be considered as a measure of nest ventilation. These methods have successfully been applied in studies on termite and rodent nests. However, we believe there are fundamental differences we should consider when working with leaf-cutting ants. The first, and most important, is that nest chambers in the *Acromyrmex* genus are

always filled with fungus garden. The only nest portions free of fungus are the tunnels connecting to the outside, which comprise a relatively low portion of the total underground nest surface. In this sense, working with an empty chamber would not recreate at all the natural conditions within a nest chamber. In the presence of a spongy-like structure as the fungus garden, the produced CO₂ remains partially and temporarily trapped inside the fungus, thus reducing the effect of any ventilating mechanism involved in the exchange of CO₂ either with the atmosphere, or with the air phase of the soil. Moreover, the concentration of CO₂ inside the chambers is not only the consequence of the fungus, yet also of the presence of the larvae and adults living inside the fungus gardens. Second, and regarding the injection of a single bolus of controlled volume, ventilation under natural conditions has also to act against a simultaneous production of CO₂, i.e., the expected production rate of the colony. We therefore preferred to study the effect of wind induced-nest ventilation under more realistic conditions by allowing both effects to be present, i.e., the effect of the fungus structure and the natural production of CO₂ at the whole colony level, including the ants.

A final comment regarding this point. The use of a fungus garden, instead of the injection of a controlled CO₂ volume as suggested by the Reviewer, was not motivated to rule out any particular hypothesis not clearly stated in the text. The use of an empty chamber as suggested by the Reviewer, either working with a controlled injection or a bolus of fixed volume, would have forced us to open a discussion and to speculate about “what would have happened in a nest chamber housing a fungus garden with its natural rate of CO₂ production”. As indicated in the previous paragraph, our approach with the use of a fungus garden allowed us to explore the effects of nest architecture and wind on CO₂ levels and ventilation under more realistic conditions

Regarding the measured data and their visualization, there are different ways in which the magnitude of nest ventilation could be represented, i.e., the analysis of either production or depletion rate, with or without the presence of wind blowing above the nest entrances. In order to simplify the representation of the results, we decided to compare the concentration of CO₂ after an hour under different wind speeds, as compared to those measured inside a tightly closed chamber with no ventilation at all. As indicated in the text, the initial CO₂ levels in all our laboratory experiments were very low, near those of the atmospheric air inside a room (ca. 0.10 %), and measurements ran over 60 minutes. Therefore, the absolute CO₂ levels plotted in our Figure 3 could in fact be represented as a rate of CO₂ increase within an hour. We agree that curves representing the change in the CO₂ values over time would be informative about the respiration physiology of the fungus and ants under the increasing CO₂ concentrations, yet our goal was not to open a discussion about this topic in the present work. We are currently performing a detailed study on respirometry of the fungus garden at different CO₂ levels, to be published in the near future. Alternatively, the magnitude of nest ventilation could also be presented as the differences in CO₂ levels between chambers exposed to wind, and the closed, control chamber, being this difference the amount of CO₂ exchanged from inside the chamber to the outside in one hour. We are of the opinion that such differences are already shown in our results, since conclusions are based on comparisons with the closed chamber for each of the three nest architectures.

We agree with the Reviewer that fixing the time of the measurements at one hour or at any other time could sound arbitrary, since in natural situations the CO₂ concentration is a dynamic value that may vary depending on several factors. Thus, the decision of the limit time was a crucial one. We fixed the time at one hour in order to avoid that the concentration inside the chamber reached harmful values for the fungus, as we reported in previous publications. Based on them, we know that workers prefer places to rear the fungus and brood, and even digging new fungus chambers, having values near or below 3% CO₂ (Römer, Bollazzi and Rocas, 2017, 2018, 2019, cited in the manuscript).

Values above this level are assumed to be harmful for the fungus and larvae. In preliminary experiments, we determined that the CO₂ concentration in the chambers increased within an hour but not surpassed the critical level of 3 %. When kept in the laboratory, the nest boxes are well ventilated because of the presence of several openings on the lid, covered by a metal mesh. During experiments, nest boxes can only be ventilated through one nest entrance in the case of the serial nest, so that they may rapidly achieve harmful CO₂ levels. Rearing leaf-cutting ant colonies in the laboratory are rather difficult, expensive and it can take years to raise an adult colony. Besides, only few species adapt well to the laboratory rearing conditions. Thus, we have set the limit of time of an hour to preserve the colonies' and the fungus gardens' health. Going beyond an hour, CO₂ production rate would be affected and measurements could not be compared because the negative effects caused by the experimentally-induced hypercapnic conditions on the fungus. We have clarified this point in the methods section (Lines 234-239 in the Methods)

Reviewer 1 _ comment 3

Regarding field measurements:

1. Fig. 4C was a little mystifying to me. The correlation between wind and CO₂ levels, as presented, appears way weaker than I would have thought, and I wonder if there was some structure in the data that was not properly accounted for when processing the data. Particularly the many points showing positive correlations between wind and CO₂ are hard to accept. How should we understand them? Over what interval in time were the wind speed and CO₂ measurements grouped? I can imagine that a wind event following some stagnation could churn up high CO₂ air from the nest toward the sensor, but the very same event should lower the CO₂ over a somewhat longer interval. Could that type of situation be reflected in the results as reported? Perhaps a different method of processing the raw data (eg. low pass filter) would extract the physically meaningful signal from high frequency noise. If another method doesn't work, it could be useful for an interested reader to see more of the raw data, not unlike that shown in Fig. 4B, in the SI.

ANSWER Reviewer 1 _ comment 3:

DONE. We agree that the correlations between wind and CO₂ levels, as presented, are weak. However, we should point out that instead of calculating a normal r-value, we have applied a repeated measures correlation. This procedure is suitable for assessing the correlation between two variables obtained by repeated measures within a subject (a nest), by estimating a common correlation value (rmcorr) that not only takes into account the non-independent repeated measurements on a nest, yet also considers the difference among subjects, i.e., the four nests in our case. If we would have applied the common correlation coefficient (r) by pooling the data of all four nests without considering their differences, correlation values would be indeed higher. For instance, compared to the rmcorr values of -0.21 and -0.26 for *A. heyeri* and *A. lundii*, values of the normal correlation coefficient (r) for *A. heyeri* would be -0.34 and for *A. lundii* -0.41. The fact that rmcorr is low indicates that the percent change of CO₂ as a consequence of wind variation is statistically significant, yet also that nests respond different to wind. We consider these results expected, taking into account that they were obtained in the field, and besides, that the CO₂ concentration was measured inside the fungus garden. As discussed above for the laboratory experiments, the spongy-like structure of the fungus garden works as a low-pass filter and is therefore expected to

preclude an immediate change in CO₂ levels after changes in wind speed, since CO₂ has to be pulled out from inside a large fungus garden, up to 50 cm in diameter (see Fig 1). Considering this, and contrary to the view of the Reviewer, we were in fact surprised that a correlation between wind speed and CO₂ levels was obtained at all, even a correlation that explains 25 % of the variation. In addition to the presence of the garden, there are other factors besides wind that contribute to changes in CO₂ levels over time, such as the soil acting as a source of CO₂ (see Discussion), as well the mass movements of air driven by convection (Fig. 5). In this context, the finding that a significant proportion of the percent change of CO₂ inside the nest correlates with variations in wind speed is a very relevant result that highlights the importance of wind-driven ventilation, despite other factors that may play a role.

In this line of arguments, it is not surprising that wind may also promote an unexpected *increase* in the CO₂ concentration inside the fungus garden under some circumstances. For instance, wind may promote a movement of air movements inside the fungus garden that shifts CO₂-rich air from the lower portion of the nest towards the fungus garden, close to the place where air is being sampled. Nevertheless, an increase in wind speed resulted, in most cases, in an expected negative change in the percent change in CO₂ as reflected in Fig 4c, and not in a positive change.

Following the Reviewer's suggestion, we used a different method for processing the raw data in order to smooth the high-frequency variations in the measured changes of CO₂ over time. We deeply thanks Reviewer 1 for such fundamental suggestion. In summary, we averaged data of all four nest pairs for time periods of 30 min, and performed all further analysis with these values. Besides, we added a Figure in the Supplementary Material (ESM 3), as suggested by the Reviewer, showing examples of the other three nests pairs measured. Details are given in the text, lines 295-301 in the Methods and 361-382 in the Results.

Reviewer 1 _ comment 4

A general issue that kept coming to mind while reading is the role of the harder-to-know rate and spatial density of CO₂ production in putting pressure on the different exchange mechanisms. In some places, "CO₂ levels" seem to take the place of the more relevant notion of rates of exchange. For instance, that "... levels of CO₂ inside fungus gardens in serial nests would equilibrate with those of the soil phase, and vary depending on the depth..." assumes production of CO₂ is very slow compared to the rate of diffusion into the soil. Comment on the validity of this assumption is warranted. Minor point: Rather than truly equilibrate, I'd say the long-term CO₂ concentration profile would reflect a 'steady-state' condition or 'dynamic equilibrium'.

In any case, it makes sense that increased size of the nest, and therefore increased rate of production of CO₂ plays a dominant role in requiring adaptation in ventilation mechanisms. Of the large serial nests (like Atta) without wind-driven exchange, according to the results, has there been analysis of the size and spacing of chambers that balance the CO₂ production rate with the diffusive exchange rate in the soil? Just a thought.

ANSWER Reviewer 1 _ comment 4

DONE. We agree with the reviewer about the importance of the rate of production of CO₂ on the effectiveness of the different ventilating mechanisms. Despite the comment is referred "just as a thought", we are of the opinion that it deserves an answer in the manuscript. When in the text we

refer to CO₂ levels inside nests, we agree that they represent in fact “snapshots” taken at a specific moment from a steady-state equilibrium, which results from the exchange rates between the fungus and the adjacent soil. We added a sentence in the discussion to explicitly introduce this concept (lines 426-427). Furthermore, the direction of the exchange would depend on the concentration of the soil and fungus relative to each other, which in fact determines the direction of the diffusion potential. When discussing in the manuscript the CO₂ levels inside serial nests and their dependence on soil levels, we stated: “Thus, levels of CO₂ inside fungus gardens in serial nests would equilibrate with those of the soil phase, and vary depending on the depth at which the chambers are placed. The deeper the nest location across the soil profile, the higher the levels, since CO₂ in soil is positively related to depth.” This in fact recognizes that there is an equilibrium with the soil. However, we agree that we did not properly discuss the notion that this equilibrium depends on the rate of production of CO₂ by fungus and soil relative to each other, and that the production rate of CO₂ by the fungus could be lower or higher than the diffusion rate into the soil. Nevertheless, we disagree with the Reviewer that we have assumed that the production of CO₂ was very slow compared to the diffusion into the adjacent soil layer. We just missed in the manuscript to emphasize this important point. We now modified the discussion to address this point, (lines 427-443), and we again thanks Reviewer 1 for the very important suggestions.

Reviewer: 2

Comments to the Author(s)

The authors perform field and laboratory assays to determine how three different nest architectural plans influence CO₂ ventilation in Acromyrmex fungus-farming ant colonies. CO₂ levels within nests were shown to be strongly dependent on nest architecture, with thatched mounds acting as superior means of increasing gas exchange. This paper will be of broad interest to readers interested in functional morphology of the “extended organism” and the evolution of unique blueprints in response to simple tradeoffs between the regulation of humidity, temperature and ventilation. The interpretation of the results is well supported by the data, and I found the figures especially appealing (figure 3, in particular). I have several suggestions that would improve this draft of the manuscript:

1. You note that serial chambers architecture is basal/more ancient, but do not mention the lower attines that build alternative nest types, which are not serial. For instance, the yeast growing Cyphomyrmex, and some Apterostigma spp. maintain shallow fungus gardens under loose leaf litter, in cavities in wood, and on in curled leaves, with a veil of fungus that acts as a “skin” over the colony and exposed garden. I agree that “ancestral ants” likely produced a simple serial nest. However, you are underrepresenting the diversity in the lower attines.

DONE. We agree and added in the Introduction a sentence on the diversity of nest architectures of all fungus-growing ants, to clarify this point (lines 89-96)

2. You state that “To our knowledge there are no studies on evolutionary trade-offs that may account for differences in architectural traits in ant nests.” However, I would argue that you and your colleagues have already demonstrated that turrets are an adaptive feature of nest architecture, in the contexts of ventilation and flooding. Several other studies of the adaptive function of chamber size and arrangement have been undertaken (Tschinkel 2018, Solenopsis invicta; Vaes et al. 2020, Myrmica ruba – though this study has design issues). Also, the function of nest mound architecture in preventing inundation by water, and funnel-entrances and narrow entrance shafts preventing invasion by army ants in Ectatomma and other ant species. In other words, I would caution you not to suggest this is the first attempt to understand evolutionary tradeoffs of nest architecture in ants.

DONE. We completely agree with the Reviewer. Our sentence was erroneously formulated, and we apologize for the mistake. The sentence was not intended to express that our work was the first that studied evolutionary trade-offs related to architectural traits in ant nests. Our intention was to emphasize that we investigated for the first time, in a comparative way, “trade-offs related to control of nest climate”, following the argumentation of the previous sentences about termites.

We have changed the sentence and referred specifically to the control of nest climate (lines 166-169 in the Introduction), and also added the suggested reference by Tschinkel 2018 (line 156).

3. Introduction page 1, Line 27 (opening sentence): Change “could” to “can”

DONE

4. You provide estimates of Atta chamber number, but roughly how many chambers or garden chambers do serial Acromyrmex colonies have in nature?

DONE. Line 108

5. Page 4, line 46: change “selected for to” to “selected to”

DONE

6. I am glad that you made your comparisons “between the CO₂ concentration for each measurement at a given wind speed minus the average concentration measured in total absence of wind for the nest under consideration.” However, your laboratory design prevents you from comparing between nest designs, because total chamber volume differs considerably between treatments. You may want to explain why you did not standardize chamber volume and make a note on figure 3 to indicate that comparisons cannot be made between nest designs.

DONE. The Reviewer’s comment appears to be a bit confusing, since the first sentence cited by the Reviewer refers to the field experiments (lines 339-341 V1 sent for Revision), and the following sentence refers to the laboratory experiments. Nevertheless, it is important to clarify the issue and provide an explanation as requested by the Reviewer. First, we indeed standardized chamber volume as explained in lines 197-199 (V1 sent for Revision). The same box size (chamber size) was used for the serial, shallow and thatched nests. We recognize that it was not properly clarified in the text, and we have improved this part in the revised version (lines 211-212 / 214 in the Methods). Thus, chamber volume was fixed and invariant for the three nest types. What differed is total nest volume between the three nest designs, as follows: 4 boxes filled with fungus gardens in the serial nest vs 1 in both the shallow and the thatched. Nevertheless, we are of the opinion that differences in nest volumes between the nest types do not invalidate our conclusions, since we did in fact not compare absolute CO₂-levels among the nest types, but *within* a nest type. We compared ventilated vs closed chambers (i.e., ventilated at wind speeds of 0, 1 and 2.5 m/s vs closed chambers), for each of the three nest architectures, as explained in lines 209-214 (V1 sent for revision). To further clarify this point, we have rewritten and extended the corresponding section in the Methods (lines 194-201 in the Methods), and therefore think that there is no need modify the caption of the Figure 3, as suggested by the Reviewer.

7. Add sample sizes to the caption of fig 3.,

DONE

8. Discussion: The following sentence about tradeoffs between nest types is key to your argument, and should be emphasized earlier in your discussion, and perhaps even the introduction: “Although a serial arrangement of nest chambers constrains nest ventilation, it provides benefits in terms of humidity and temperature regulation, because soil humidity increases with depth [59], and suitable soil temperatures for fungus rearing are found in the deep layers of tropical soils instead of at the surface level [29].” Unless you very clearly describe these tradeoffs early on, readers will question

your assertion about adaptive architecture, given that Atta colonies achieve great success with a serial nest design.

DONE: We completely agree, many thanks for this hint. We had in fact mentioned possible trade-offs between thermoregulation and gas exchanges in the Introduction of the original manuscript, but in general terms, referring to examples of termites. In the Introduction of the present revised version, we have now explicitly referred to these trade-offs in relation to the different nest architectures observed in leaf-cutting ants, to emphasize our key arguments (lines 152-154 / 169-175).

9. The discussion could be streamlined and shortened.

Given that Reviewer 1 asked for a clarification of some points we agree they deserved an answer, it was not possible to substantially shorten the Discussion.

10. I enjoyed the supplementary videos.

THANKS.

Finally, we have thoroughly revised the text for proper English usage and improved a number of sentences.

Appendix B

Editorial Office of Royal Society Open Science

Montevideo, Uruguay / Würzburg, Germany, 21 September 2021

Dear Editors of Royal Society Open Science!

We would like to submit the REVISED version of our manuscript (RSOS-210907.R1) entitled **“Carbon dioxide levels and ventilation in *Acromyrmex* nests: significance and evolution of architectural innovations in leaf-cutting ants”**, by Martin Bollazzi, Daniela Römer and Flavio Roces, to be considered for publication in *Royal Society Open Science*.

We are very much indebted to the Reviewer 1, whose comments were very helpful to improve the revised version of our manuscript. We are providing a revised version in which we addressed all the Reviewer queries without increasing the overall size of the manuscript. All replies to the Reviewer 1 questions can be found in the ‘response to reviewers’ word file (see bellow) we uploaded to the Royal Society editorial manager.

We hope you will agree with our detailed answers and the corresponding changes in the text, when giving our manuscript further consideration.

Thank you again for considering our manuscript for publication in Royal Society Open Science!

sincerely yours,

Dr. Martin Bollazzi, Dra. Daniela Römer and Dr. Flavio Roces

Associate Editor Comments to Author:

Comments to the Author:

Please see the remaining comments from Reviewer 1. While a further review process is unlikely, this is also contingent on your satisfying the Editors that you've taken the reviewer's concerns seriously and addressed them not only in the response to reviewers but the manuscript itself (please ensure a tracked-changes version of the - hopefully - final iteration of the paper is included when you resubmit). The reviewer also poses a question about the availability of data: in the event you have data regarding this point, please ensure it is made accessible in-line with the journal's open data policy (and check with the editorial office if anything is unclear here); if you do not have direct access to data, but can point to a supporting reference or two, this would be helpful.

Answer _ Associate Editor

DONE: thanks for your helpful guidance. We have now read and carefully responded to the remaining comments raised by Reviewer 1, both in the text below as well as in the revised manuscript version R2.

Reviewer comments to Author:

Reviewer: 1

Comments to the Author(s)

I appreciate the authors' work in revising the manuscript. I do have some remaining concerns that I hope can be addressed without major changes:

Reviewer 1 _ comment 1

1. The role of soil porosity in wind-driven ventilation. This may be addressed in words without much additional work, but will serve to justify the wind experiments. I apologize if I had not effectively expressed my question before. If you suck on a straw stuck deep in the sand, you can pull some air out, because the permeability of the sand integrated over a large area underground gives some less-than-infinite resistance. If the straw ends in an impermeable chamber, you can't drive flow. This is why I had insisted that the potential effect of the wind in driving some ventilation necessarily depends on the porosity of the soil, in principle. If it is hard to force air into or out of the entrance holes of the nests in the field, it would indicate my point is probably not practically relevant. I wouldn't know which to guess, but it changes the significance of conclusions drawn from the lab experiments.

Response Reviewer 1 _ comment 1:

DONE: We thanks again Reviewer 1 for his/her deep thoughts about the mechanisms behind nest ventilation, which we consider have enriched our work. Regarding soil porosity, we believe there are

two levels of consideration we should address independently. First, the general background of mechanisms acting on the study systems, and second, the methodological approach. Regarding the general background, surface winds may potentially draw underground gases through a single nest opening in a porous soil, yet such flows would draw CO₂-rich soil air from the soil into the nest environment. As a consequence, no gas exchanges with the free atmosphere (ventilation) would occur in serial nests with a single opening. Therefore, the potential occurrence of diffusive flows in such nests does not change the conclusions drawn from the laboratory experiments. Regarding the latter, we should emphasize that the experiments focusing on wind-induced flows were intended to study the pure effect of nest design on ventilation via the nest tunnels alone, precluding the effect of other variables such as temperature and soil porosity, as already indicated in the first revised version (line 189 R1). We had in fact justified the use of totally impermeable plastic boxes to preclude any exchange of gases through the chamber walls (lines 226-229 R1). Nevertheless, we agree with the Reviewer that the effect of soil porosity on gas exchanges between the soil and the colony deserves further consideration in the general background of the study, and we have proceeded as follows. Right at the beginning of the Introduction (5th line of the first revised version, line 43 R1), we had already recognized that nests exchange respiratory gases with the air phase of the soil via diffusive flows. However, we have now extended the text by stating that soil porosity would in fact determine the extent of such exchange (lines 44–45 and 49-50 R2). In addition, we have now explicitly addressed the effect of soil porosity in the way suggested by the Reviewer in the Discussion (lines 494 – 500 R2), because we agree that this topic deserves more attention. However, we would like to emphasize that we had already discussed it in the R1 (lines 437-439). We thank again Reviewer 1 for this important hint.

Reviewer 1_ comment 2

2. The direction of net flux of CO₂ underground. Between my first reading of the manuscript and the revision, my confusion has shifted. What I would like to visualize better is a physically consistent picture of how CO₂ gets from source to atmosphere. As noted in the manuscript, the soil itself is producing CO₂, and due to this production, the CO₂ levels are known to increase with depth -- and this is important for the issue of transport between nest and soil. However, the more specific nature of this balance is unclear.

I want to be careful about my terms here:

Equilibrium = no concentration difference between 2 bodies, therefore no transport between them (via Fick's law)

Steady state = condition in which the net flux of CO₂ out of any element is equal to its rate of production, such that the level measured at any location is constant over time (it isn't accumulating anywhere at steady state)

I guess "dynamic equilibrium" is a valid equivalent to "steady state" above.

The added lines 427-443 confused me here. The suggestion that CO₂ diffuses into the nest at larger depths leads to a couple of different conclusions which seem important and counterintuitive.

(At first reading, one might interpret the nest metabolism runs backwards to accommodate the boundary conditions...) The direction of flux between nest and soil should be determined by the local rate of CO₂ production, not the CO₂ levels in soil without the nest. Do we assume that the soil at deeper depths produces more CO₂ volumetrically, or are the observed values representative of the steady state of constant flux toward the surface of CO₂ produced at constant rate at different levels? If the latter, for instance, CO₂ will still flow from nest to soil (because the production rate is

volumetrically greater in the nest), but the steady state concentration of the nest will simply surpass the ambient value at that depth. The former is really hard to imagine, but would necessarily mean that for every bit of flux into the nest from the neighboring soil (laterally), there must be that much more flux outward (presumably toward the surface) to be at steady state. Further comment and clarity here would be very helpful. (Analysis of a similar situation in Ref. 62 (Wilson, Kilgore 1978) was useful for me.)

Response Reviewer 1 _ comment 2:

We thank again Reviewer 1 for this comment and for having done a great and insightful elaboration on the dynamic of mass movements of CO₂ between soil and nest. We agree that having a better picture of how CO₂ gets from source (a fungus chamber in a serial nest) into the atmosphere will be desirable. We would like to mention that we had already addressed such topic in lines 508-510 R1, but we agree it was a short elaboration and now we have improved such part in this revision following the Reviewer's suggestion (see last paragraph at the end of this section for lines details in R2).

In the original version and R1, we have explained that our laboratory experiments were aimed at answering the question about the relevance of architectural features (arrangement of tunnels, presence of a nest thatch) for nest ventilation in a comparative way, considering the three different nest designs we identified in the genus *Acromyrmex*. Our experiments were not aimed at investigating the physics of gas exchanges in the system soil-nest, which are complex and dependent of a number of variables beyond the sole rate of CO₂ production (such as temperature, daily thermal cycles, soil moisture, soil granulometry, etc.). Nevertheless, we have discussed in the actual revised version what CO₂ concentrations fungus gardens inside serial nests will be exposed to in the long term. We stated that in serial nests, CO₂ concentration inside fungus chambers is expected to depend on the depth at which the chambers are placed (lines 430-437 and 503-509 R2), as we also demonstrated in a previous publication on *Atta* nests (reference 3), and because of the lack of wind-driven ventilatory flows (see our comment above), it will tend to equilibrate with that of the soil.

Reviewer 1 claims that he/she was confused about our suggestion that CO₂ diffuses from the soil phase into the nest at larger depths, and added: "the direction of flux between nest and soil should be determined by the local rate of CO₂ production, not the CO₂ levels in soil without nest". We agree that the direction of flux depends on the rate of production of both, soil and fungus gardens, and we should emphasize we had already commented on this using almost the same wording in lines 44-48 R1 and 427-429 of R1. Nevertheless, we agree it deserves further explanations and we have proceeded as requested (see below, at the end of this section, for lines details in R2).

Whatever the mechanisms of CO₂ exchange between the air phase of the soil and the fungus gardens, the CO₂ concentration inside the chambers of a serial nests will tend to equilibrate with that of the soil in the long term, since no ventilatory airflows through the single nest opening occur. This is the main conclusion we have formulated regarding serial nests. When compared with the two other nest designs, while serial nests cannot capitalize on wind-induced ventilatory flows, both shallow and thatched nests benefit from wind-induced nest ventilation via nest tunnels.

We have changed the discussion considering the Reviewer's request, using so few words as possible. To address the 2nd comment of Reviewer 1, we proceeded as follows. (I) We have emphasized that the direction of CO₂-flux between fungus and soil in serial nests may vary depending

on the actual rates of production (lines 497-500 R2); (II) we have emphasized that in the serial nests, concentrations on the two phases may tend to equilibrate in the long term given the absence of direct air exchanges with the atmosphere (line 435-437 and 497-498 R2); (III) we have explained that surface winds, despite not inducing circulation of atmospheric air via tunnels in serial nests, can draw air from the soil into the chamber through the chamber's walls, highlighting the importance of soil porosity (the "straw metaphor" in the Reviewer's words) (lines 494-497 R2); (IV) we have further explained that the long-term dynamics of CO₂ inside fungus gardens and soil are linked (lines 498-500 R2); and (V) we have argued that nocturnal thermal convection is an additional driving force of air exchanges with the atmosphere, which distinctly influences chambers located at different depths (lines 503-509 R2).

In summary, we have rearranged and modified the sections of R1 where we had summarized the involved ventilation mechanisms acting on serial nests (now in lines 417-437 R2 and 488-511 R2), besides the other two nest types (lines 520-525 R2). It was done without increasing manuscript length, since R1 had 604 lines and this new R2 have 610 lines. We now believe we have satisfactorily addressed the request of the Reviewer of "better visualize a physically consistent picture of how CO₂ gets from source [fungus chambers] to atmosphere". We thank Reviewer 1 again for helping us to better direct our discussion regarding the mechanisms of nest ventilation.

Reviewer 1 _ minor point

Minor point:

"We agree that curves representing the change in the CO₂ values over time would be informative about the respiration physiology of the fungus and ants under the increasing CO₂ concentrations, yet our goal was not to open a discussion about this topic in the present work."

--The discussion which is open here does not just concern respiration physiology. How the CO₂ levels approach the reported values over time is relevant to a reader interested in the mass transfer dynamics (arguably the basic substance of this work). If the data is not available for this paper, that is ok, but I don't agree with the rationale for its omission.

Response Reviewer 1 _ minor point:

Unfortunately we do not have the curves representing the rates of CO₂ change. We agree that a reader interested in mass transfer dynamics would benefit from seeing the slopes of the curves, yet we had already knew that the increase of CO₂ over 1 hour is linear. From the publication by Kleineidam and Roces 2000 (reference 4), we know that the production of CO₂ by an enclosed fungus garden of the leaf-cutting ant *Atta sexdens* (which cultivates the same fungus species of *A. lundii*, reference 59) follows a linear increase up to approximately 3 % over one hour, and this is also the reason why we decided to use such time interval for our measurements. We are of the opinion that knowledge of the dynamics of CO₂ changes over time, which is actually lacking, would not change the conclusions of our comparative results among the three different nest architectures. Such knowledge would for sure enrich the discussion about the underlying ventilation mechanisms, and we are currently working, as mentioned in our previous reply, on a follow-up study in which we investigate the dynamics of CO₂ production over time and its effects on both fungus and ant respiration.

As requested by Reviewer 1 and by the Editor we have added a text portion in the methods explaining all concepts detailed above, as well as the corresponding reference [4] (lines 240-242 R2).

Reviewer: 2

Comments to the Author(s)

The authors have significantly remodeled the manuscript in response to both reviewers -including clarification of background, additional details

Response to Reviewer 2:

THANKS for your helpful work throughout the whole review process.